# An observation-based evaluation and ranking of historical Earth System Model simulations in the northwest North Atlantic Ocean

**Arnaud Laurent[1], Katja Fennel[1], and Angela Kuhn[2]**

[1]1Department of Oceanography, Dalhousie University, Halifax, Nova Scotia, Canada
[2]Scripps Institution of Oceanography, UC San Diego, USA

**Correspondence:** Arnaud Laurent (arnaud.laurent@dal.ca)

**Abstract.** Continental shelf regions in the ocean play an important role in the global cycling of carbon and nutrients but their responses to global change are understudied. Global Earth System Models (ESM), as essential tools for building understanding of ocean biogeochemistry, are used extensively and routinely for projections of future climate states; however, their relatively coarse spatial resolution is likely not appropriate for accurately representing the complex patterns of circulation and elemental fluxes on the shelves along ocean margins. Here, we compared 29 ESMs used in the IPCC's Assessment Rounds (AR) 5 and 6 and a regional biogeochemical model for the northwest North Atlantic (NWA) shelf to assess their ability to reproduce surface observations of temperature, salinity, nitrate, and chlorophyll. The NWA region is biologically productive, influenced by the large-scale Gulf Stream and Labrador Current systems, and particularly sensitive to climatically induced changes in large-scale circulation. Most ESMs compare relatively poorly to observed surface nitrate and chlorophyll and show differences with observed surface temperature and salinity that suggest spatial mismatches in their large-scale current systems. Model-simulated nitrate and chlorophyll compare better with available observations in AR6 than in AR5, but none of the models performs equally well for all 4 parameters. The ensemble means of all ESMs, and of the five best performing ESMs, strongly underestimate observed chlorophyll and nitrate. The regional model has a much higher spatial resolution and reproduces the observations significantly better than any of the ESMs. It also simulates reasonably well vertically resolved observations from gliders and bi-monthly ship-based monitoring observations. A ranking of the ESMs indicates that only 1 ESM has good and consistent performances for all variables. An additional evaluation of the ESMs along the regional model boundaries shows larger variability but is generally consistent with the ranking on the shelf. Overall, 11 ESMs were deemed satisfactory for use in the NWA, either directly or for regional downscaling.

## 1  Introduction

Elemental fluxes along ocean margins, which are areas of complex physical and biogeochemical interactions, are important components of the global cycles of carbon (C) and nitrogen (N). For example, continental shelves host up to a third of oceanic primary production and over 40% of carbon burial in the ocean (Ducklow and McCallister, 2004; Muller-Karger, 2005; Walsh, 1991). They also are important sites of sediment denitrification leading to a net removal of fixed nitrogen (Fennel et al., 2006; Seitzinger and Giblin, 1996). Many shelf regions are thought to be a significant sink for atmospheric $CO_2$ (Cai et al., 2006; Chen et al., 2013; Laruelle et al., 2018), including the eastern margin of North America (Fennel et al., 2019, and references therein), although there are significant discrepancies in available estimates. Despite their importance, the response of ocean margins to climate change is understudied relative to the open ocean.

Future projections of ocean biogeochemistry rely heavily on Earth System Models (ESMs). These are state-of-the-art, comprehensive representations of the major earth system components (including atmosphere, ocean, and land surface) and are routinely used to perform climate scenario projections. The spatial resolution of the CMIP-class ESMs typically ranges from 0.5 to 2° and is too coarse to resolve coastal ocean dynamics and interactions between shelf and the open ocean (Anav et al., 2013; Bonan and Doney, 2018; Holt et al., 2017). This leads to uncertainty in future projections, not only for margin regions, and a global underestimation of

the high primary productivity in coastal regions (Bopp et al., 2013; Schneider et al., 2008).

Regional coupled circulation-biogeochemical models have been developed at much higher spatial resolution. These regional models have been used to investigate biogeochemical processes along ocean margins (Fennel et al., 2006, 2013; Lachkar and Gruber, 2011; Peña et al., 2019; Siedlecki et al., 2015; Zhang et al., 2020) and project future states resulting from climate change (Gruber et al., 2012; Hermann et al., 2016; Holt et al., 2016; Laurent et al., 2018). The regional models allow for the temporal and spatial resolution necessary to resolve mesoscale processes and can be regionally calibrated (e.g., Kuhn and Fennel, 2019; Mattern and Edwards, 2017). However, the dynamics of a regional model is strongly determined by information imposed along the model's open lateral boundaries, typically derived from a larger scale model, reanalysis product, or observation-based climatology. For future climate simulations, a regional model requires boundary information from future projections of large-scale models or ESMs.

The northwest North Atlantic (NWA), located at the confluence of the subtropical and subpolar gyres, is particularly challenging to global ocean circulation models and highly sensitive to climate-induced modifications of the large-scale circulation, which are thought to be responsible for a multi-decadal deoxygenation trend in the region (Claret et al., 2018; Gilbert et al., 2005, 2010). While the CMIP models reasonably describe the large-scale climatological features of ocean physics in the NWA, the detailed current structure is poorly represented due to a mismatch in the location of the subtropical and subpolar gyres (Loder et al., 2015). The Gulf Stream usually extends too far north and the branch of the Labrador Current flowing southwest along the shelf edge tends to be missing (Lavoie et al., 2019; Loder et al., 2015). This leads to a warm bias in the NWA, a common feature among coarse resolution ESMs (Saba et al., 2016). The absence of the shelf-break current significantly impacts cross-shelf exchange with much larger shelf water residence times in a high-resolution regional model (Rutherford and Fennel, 2018) compared to estimates from a global model (Bourgeois et al., 2016). These discrepancies have been attributed to the coarse resolution of the global models (Lavoie et al., 2019; Loder et al., 2015; Rutherford and Fennel, 2018; Saba et al., 2016). Despite these issues, CMIP historical simulations and future projections have been used to characterize biological responses to climate change in the NWA (e.g., Bryndum-Buchholz et al., 2020a; Greenan et al., 2019; Lavoie et al., 2019; Stortini et al., 2015; Wilson et al., 2019; Wilson and Lotze, 2019). ESM selection in these regional studies is either qualitative or based on either scenario outcomes (e.g. variability across models) or global assessments rather than on regional model performance. However, ESMs that poorly represent the dynamics of the NWA will affect the results of regional studies.

Increased coastal model resolution can be achieved by down-scaling large-scale or global models, the so-called parent models, to high-resolution regional models, the child models (see, e.g. Hermann et al., 2019; Holt et al., 2016; Laurent et al., 2018; Lavoie et al., 2020). For future projections, the obvious approach is to downscale ESMs. Since simulation of the fine-scale processes in the child model is strongly influenced by the parent model, it is important to assess the skill of ESMs in reproducing historical observations prior to using them for downscaled future projections. Rickard et al. (2016) ranked ESMs based on their misfit with regional observations around New Zealand in order to discard models with significant errors and determine an ensemble of "best" models that can be used to study regional climate projections, either directly or indirectly through regional downscaling. Here, we take a similar approach.

Our main objective is to assess the performance of a number of available ESMs in reproducing present conditions on the NWA shelf in contrast to a high-resolution regional model. This is an important information for users of historical and future projections in the region. Additionally, we assess ESMs performance along the boundaries of the regional model. This information is necessary when downscaling with a regional model. More specifically, we compare 29 ESMs used in the two most recent IPCC Assessment Rounds (AR) as part of the Coupled Model Intercomparison Project 5 (CMIP5; Taylor et al., 2012) and its currently ongoing successor CMIP6 (Eyring et al., 2016). We carry out a systematic and quantitative assessment and ranking by comparing the CMIP5 and CMIP6 models against observed surface temperature, salinity, chlorophyll, and nitrate and perform the same comparisons for a regional biogeochemical model. The latter is the Atlantic Canada Model (ACM, Brennan et al., 2016; Rutherford and Fennel, 2018) with biogeochemistry (Bianucci et al., 2016; Kuhn and Fennel, 2019) and is intended for regional downscaling of ESM simulations in order to generate high-resolution future projections. For all models, we present statistical metrics based on the mismatch of each model with climatological surface observations of temperature, salinity, nitrate, and chlorophyll and a ranking based on these metrics. The regional model is further evaluated against in-situ measurements, including high-resolution cross-shelf glider transects. The comparison provides an overview of ESM performance in the NWA and shows sufficient confidence for only a third of the ESMs. The regional model clearly outperformed all the global models and regional downscaling using single ESM forcing (as opposed to an ensemble) is recommended.

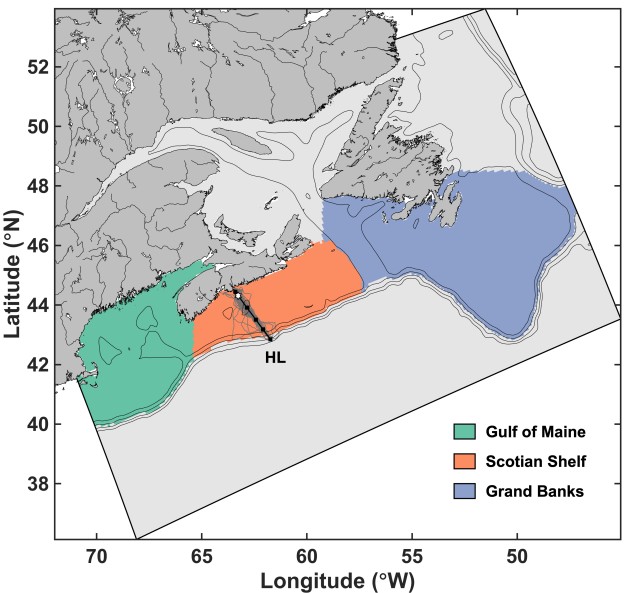

**Figure 1.** Study area indicating the 3 averaging zones, the limits of the ROMS grid and the location of the Halifax Line stations (squares) used in the analysis. The white star is Station 2 and the grey lines the gliders track.

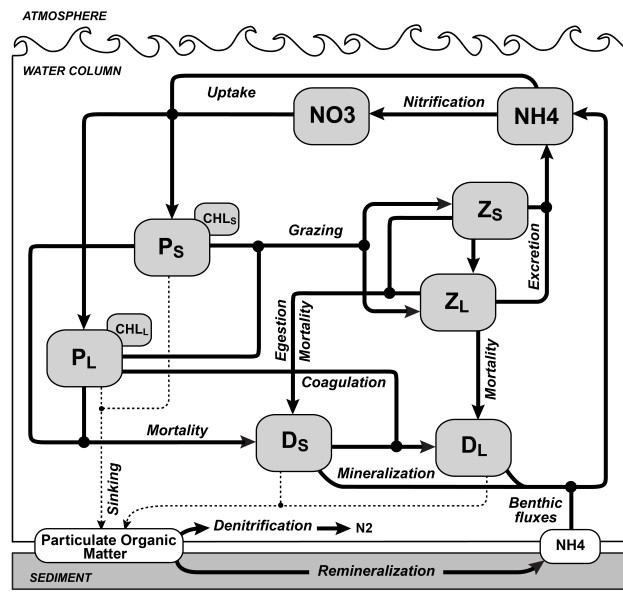

**Figure 2.** Schematic of the biogeochemical model used in ROMS. The state variables are small phytoplankton ($P_S$) and chlorophyll ($CHL_S$), large phytoplankton ($P_L$) and chlorophyll ($CHL_L$), small zooplankton ($Z_S$), large zooplankton ($Z_L$), slow-sinking small detritus ($D_S$), fast-sinking large detritus ($D_L$), nitrate (NO3), and ammonium (NH4). Dashed lines indicate sinking. Black dots represent the connections between paths.

## 2 Material and Methods

### 2.1 Models

#### 2.1.1 Global models

The CMIP5 and CMIP6 framework provides state-of-the-art climate model datasets from the previous (AR5) and current (AR6) IPCC Assessment Rounds (Eyring et al., 2016; Taylor et al., 2012). Of all the ESMs, those that include ocean biogeochemistry with monthly outputs of surface temperature, salinity, chlorophyll and nitrate were included in our comparison. A total of 29 such ESMs were available (Table 1), 17 from CMIP5 (models 2–18, downloaded from the Earth System Grid Federation (ESGF) data repository at https://esgf-node.llnl.gov/search/cmip5/) and 29 from CMIP6 (models 19–30, downloaded from the ESGF data repository at https://esgf-node.llnl.gov/search/cmip6/). These models vary in their horizontal and vertical resolution and include a total of 13 different ocean biogeochemical models of varying levels of complexity (Table 1 and references therein).

We accessed the historical simulations which were forced by observed atmospheric composition and land cover changes over the period ~1850–2005 (CMIP5) and ~1850–2014 (CMIP6). Monthly, spatially resolved climatologies of surface chlorophyll, nitrate, temperature and salinity were calculated over 30 years (1975–2005) from each ESM historical simulation.

#### 2.1.2 Regional model

The ACM is a high-resolution, regional configuration of the Regional Ocean Modeling System (ROMS, version 3.5; Haidvogel et al., 2008) for the NWA, nested within the larger ocean-ice model of Urrego-Blanco and Sheng (2012), that includes the Gulf of Maine, Scotian Shelf and Grand Banks (Figure 1). The coupled physical-biogeochemical model has 30 vertical layers and an average horizontal resolution of 9.5 km on the shelf (Table 1). Detailed descriptions and physical model validation are presented in Brennan et al. (2016) and Rutherford and Fennel (2018). The biogeochemical model is based on Fennel et al. (2006, 2008) but was expanded by splitting phytoplankton and zooplankton state variables into size-based functional groups, i.e. nano-microphytoplankton and micro-meso-zooplankton. The model was also modified by including temperature-dependent biological rates for nutrient uptake, phytoplankton and zooplankton mortality, grazing and zooplankton egestion and excretion (see supporting text). The model has 10 state variables: nitrate, ammonium, and two size classes each for phytoplankton, chlorophyll, zooplankton and detritus (Figure 2). This ecosystem structure is of intermediate complexity similar to the model of Aumont et al. (2015), which is used in 6 of the ESMs included in our study. Model parameters were optimized by Kuhn (2017) and are listed in supporting Table S1. The model description and equations are available in the

Supporting Information.

Initial and open boundary conditions for nitrate (NO$_3$) were defined from a monthly climatology (Kuhn, 2017) based on in-situ observations and the World Ocean Atlas 2009 (Garcia et al., 2010). Other biological variables were set to 0.1 mmol N m$^{-3}$ with a phytoplankton-to-chlorophyll ratio of 0.76 mmol N (mg Chl)$^{-1}$ (Bianucci et al., 2016). The model was initialized on January 1, 1999 and run through December 31, 2014. The first year was considered spin up. Monthly climatologies of surface chlorophyll, nitrate, and temperature were calculated for comparison with the ESMs.

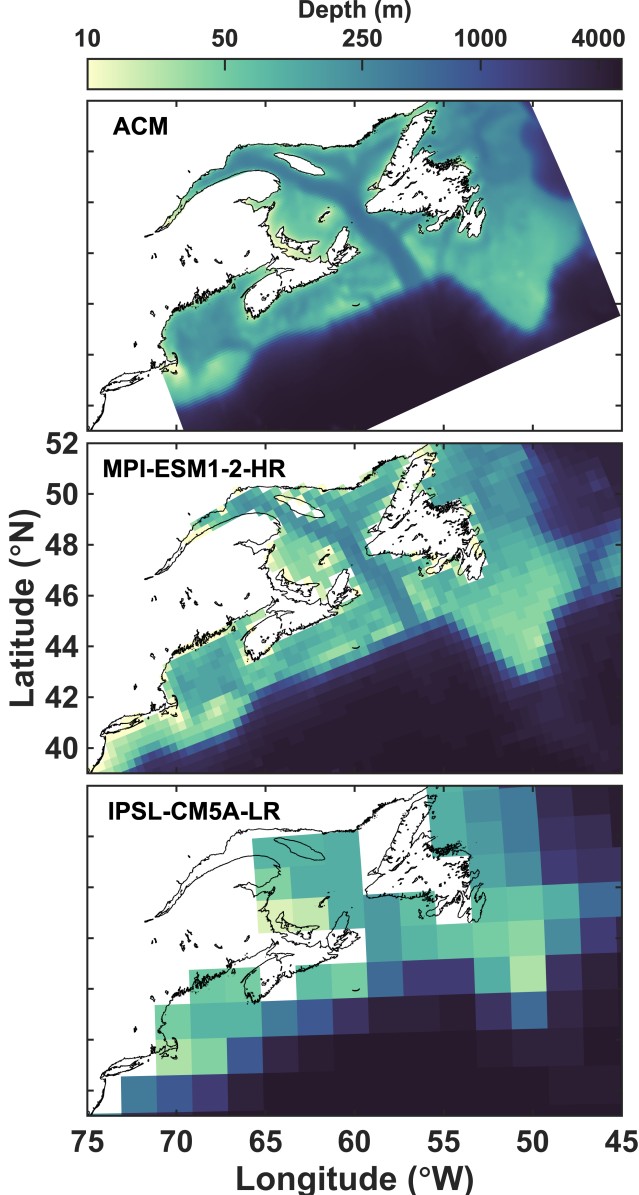

**Figure 3.** Bathymetry of the regional model (top), the highest resolution ESM (middle) and lowest resolution ESM (bottom).

### 2.1.3   Model resolution

The 30 models differ dramatically in their horizontal resolution and do not evenly cover the 3 regions of interest (Figure 3, Table 1). The regional ACM has a much higher resolution than any of the ESMs with about 16 times more horizontal grid cells than the highest resolution ESM and almost 300 times more than the lowest resolution ESM. Among the ESMs the highest resolution is achieved by models 16 and 28, which share the same grid. These two have more than twice the number of horizontal grid cells than the next highest resolution models (3, 18, 20–21). The lowest resolution ESMs are models 3 and 12–14 with only 26 horizontal grid cells within the NWA shelf resulting in a coarse representation, particularly in the SS region. The median number of grid cells in the NWA shelf region is 72 and 102 for the CMIP5 and CMIP6 models, respectively, compared to 6875 in the ACM.

### 2.1.4   Observations

Four types of observations were used in the model intercomparison: 1) satellite surface chlorophyll observations from the Sea-viewing Wide Field-of-view Sensor (SeaWiFS) as 8-day averaged maps at 1/12° resolution (1999–2010, https://doi.org/data/10.5067/ORBVIEW-2/SEAWIFS/L3M/CHL/2018), 2) surface nitrate from the World Ocean Atlas 2013 (WOA; Garcia et al., 2014)(WOA; Garcia et al., 2014) at 1° resolution, 3) daily surface temperature from the Operational SST and Sea Ice Analysis (OSTIA) system (Donlon et al., 2012) at 1/20° resolution (2006–2016, https://doi.org/10.5067/GHOST-4FK01) and 4) surface salinity from the WOA at 1/4° resolution (Zweng et al., 2013). Monthly climatologies were calculated for each of these.

In addition, the regional model was validated using high-resolution in-situ observations along the Halifax Line (Figure 1) from the Atlantic Zone Monitoring Program (AZMP, 2000–2014, http://www.meds-sdmm.dfo-mpo.gc.ca/isdm-gdsi/azmp-pmza/index-eng.html) and glider transects between 2011 and 2016 (Ross et al., 2017). To enable a quantitative comparison between the glider and ACM data (Table 3), we spatially interpolated both datasets onto a transect following the Halifax Line (black line in Figure 1). Glider missions were seasonal and therefore both glider and AZMP transects data were seasonally averaged. For each mission, data were extracted at Station 2 to produce a monthly climatology.

### 2.1.5   Comparison metrics

For comparison with the observations, each model was mapped onto the SeaWiFS, WOA and OSTIA grids using a nearest neighbor interpolation. Since some areas, such as the nearshore and the Bay of Fundy, are covered by only a

**Table 1.** Information about the regional model and the 29 ESM models. For the CMIP5 models (2–18) the r1i1p1 ensemble was used. For the CMIP6 model (19–30) the r1i1p1f1 ensemble was used on the native grid when available, except for CNRM-ESM2-1, MIROC-ES2L and UKESM1-0-LL (r1i1p1f2), GFDL-ESM4 and NorESM2-LM (regridded), and GISS-E2-1-G (r101i1p1f1). The filled circles and open squares indicate the models that are part of the inner and outer ensembles, respectively. N indicates the number of vertical levels. Note that the IPSL-CM5 models share the same ocean component with higher resolution atmospheric component in the MR version. Similarly, MPI-ESM-MR and MPI-ESM1-2-HR share the same ocean component with higher resolution atmospheric component in the HR version.

| Model | | | Shelf resolution | | | | Ocean BGC component | References |
|---|---|---|---|---|---|---|---|---|
| | | | (n cells) | | | $\Delta$lon$\times\Delta$lat (degree) | N | | |
| Name | ID | $\in$ | GoM | SS | GB | | | | |
| ACM | 1 | – | 1780 | 1366 | 3729 | 0.06×0.09 | 30 | BIO_FENNEL | Brennan et al. (2016); Fennel et al. (2006) |
| CanESM2 | 2 | □ | 11 | 14 | 29 | 1.4×0.9 | 40 | CMOC | Arora et al. (2011); Christian et al. (2010) |
| CESM1-BGC | 3 | □ | 41 | 33 | 91 | 1.1×0.4 | 60 | BEC | Lindsay et al. (2014); Moore et al. (2013) |
| CMCC-CESM | 4 | □ | 8 | 5 | 13 | 2×1.25 | 30 | PELAGOS | Vichi et al. (2007a,b, 2011) |
| CNRM-CM5 | 5 | □ | 27 | 20 | 55 | 1×0.62 | 42 | PISCES | Aumont and Bopp (2006); Voldoire et al. (2013) |
| GFDL-ESM2-G | 6 | ● | 20 | 15 | 39 | 1×1 | 50 | TOPAZ2 | Dunne et al. (2012, 2013); Dunne (2013) |
| GFDL-ESM2-M | 7 | □ | | | | | | | |
| GISS-E2-H-CC | 8 | □ | 19 | 14 | 39 | 1×1 | 26 | NOBM | Romanou et al. (2013); Schmidt et al. (2014) |
| GISS-E2-R-CC | 9 | □ | 15 | 12 | 29 | 1.25×1 | 32 | | |
| HadGEM2-CC | 10 | ● | 18 | 15 | 39 | 1×1 | 40 | Diat-HadOCC | Collins et al. (2011); Palmer and Totterdell (2001) |
| HadGEM2-ES | 11 | □ | | | | | | | |
| IPSL-CM5A-LR | 12 | ● | 8 | 5 | 13 | 2×1.25 | 31 | PISCES | Aumont and Bopp (2006); Dufresne et al. (2013) |
| IPSL-CM5A-MR | 13 | ● | | | | | | | |
| IPSL-CM5B-LR | 14 | □ | | | | | | | |
| MPI-ESM-LR | 15 | □ | 23 | 23 | 73 | 0.8×0.5 | 47 | HAMOCC 5.2 | Giorgetta et al. (2013); Ilyina et al. (2013) |
| MPI-ESM-MR | 16 | ● | 136 | 87 | 193 | 0.4×0.3 | 95 | | |
| MRI-ESM1 | 17 | □ | 40 | 29 | 80 | 1×0.5 | 50 | MRI.COM3 | Adachi et al. (2013) |
| NorESM1-ME | 18 | □ | 41 | 33 | 91 | 1×0.43 | 53 | HAMOCC 5.1 | Tjiputra et al. (2013) |
| CanESM5 | 19 | □ | 27 | 20 | 55 | 1×0.62 | 45 | CMOC | Swart et al. (2019) |
| CESM2 | 20 | □ | 41 | 33 | 91 | 1×0.43 | 60 | MARBL | Danabasoglu et al. (2020) |
| CESM2-WACCM | 21 | □ | | | | | | | |
| CNRM-ESM2-1 | 22 | ● | 27 | 20 | 55 | 1×0.62 | 75 | PISCES | Aumont et al. (2015); Séférian et al. (2019) |
| GFDL-ESM4 | 23 | □ | 20 | 15 | 39 | 1×1 | 75 | COBALTv2 | Stock et al. (2020) |
| GISS-E2-1-G | 24 | □ | 15 | 12 | 29 | 1.25×1 | 40 | NOBM | Rousseaux and Gregg (2015) |
| GISS-E2-1-G-CC | 25 | ● | | | | | | | |
| IPSL-CM6A-LR | 26 | ● | 27 | 20 | 55 | 1×0.62 | 75 | PISCES | Aumont et al. (2015); Boucher et al. (2020) |
| MIROC-ES2L | 27 | ● | 20 | 18 | 43 | 1×0.77 | 62 | OECO2 | Hajima et al. (2020) |
| MPI-ESM1-2-HR | 28 | ● | 136 | 87 | 193 | 0.4×0.3 | 95 | HAMOCC | Müller et al. (2018) |
| NorESM2-LM | 29 | □ | 25 | 20 | 57 | 1×0.6 | 70 | HAMOCC | Müller et al. (2018) |
| UKESM1-0-LL | 30 | ● | 27 | 20 | 55 | 1×0.62 | 75 | MEDUSA2 | Sellar et al. (2019); Yool et al. (2013) |

few models, grid cells that are active in less than 85% of all models were excluded from the analysis to avoid biases. In the low-resolution WOA climatology, the months November to January were excluded because poor data availability in these months resulted in unrealistic patterns.

Three zones were defined for a high-level comparison with the observations on the shelf: the Gulf of Maine (GoM), Scotian Shelf (SS), and Grand Banks (GB) (Figure 1). Subsequently, the term NWA shelf refers to the region covered by all 3 zones (GoM, SS and GB). An additional zone was also defined for a high-level comparison with the observations along the open boundaries of the ACM. Following the method of Rickard et al. (2016), a score S was calculated for each model variable, $v$ (i.e., surface tem-

perature, chlorophyll, and nitrate), for each month, t, in the climatology as the sum of the centered Root Mean Square Difference (RMSD) and bias between the observations (x) and the model (y), such that:

$$
S(t,v) = \sqrt{\frac{1}{n}\sum_{i=1}^{n}\left((x_i(t,v) - \overline{x}(t,v)) - (y_i(t,v) - \overline{y}(t,v))\right)^2} \\
+ \frac{1}{n}\left|\sum_{i=1}^{n}(x_i(t,v) - \overline{y}_i(t,v))\right|
$$

where the index i refers to a grid cell and n is the total number of grid cells within the NWA shelf. The lower the score the better the match between model and observations. An-

nual mean scores $\overline{S}(\upsilon)$ were calculated for each model vari-
able by averaging over $t$. For each variable, the models were
ranked based on their annual mean score. The overall rank
was determined by ranking models by the averages of their
ranks for surface temperature, salinity, chlorophyll, and ni-
trate ($\overline{R}$). For models with equal averages the ranking was
determined by the average of chlorophyll and nitrate ranks
($\overline{R}_{bio}$).

To facilitate the comparison with observations, the ESMs
were grouped into CMIP5 and CMIP6 and the ensemble
means of all models and of the 5 highest ranked models were
calculated for each group.

## 3  Results

Models and model ensembles are first compared with obser-
vations to assess their ability to reproduce the annual cycles
of surface temperature, salinity, chlorophyll and nitrate in the
NWA region. Error statistics are then analyzed to understand
how the models deviate from each observed variable and sub-
sequently used to calculate the scores and then rank the mod-
els. Finally, additional, high-resolution comparisons between
models and observations are presented to further assess the
regional model's performance.

### 3.1  Model-data comparisons

First, we compare the spatially averaged climatological sur-
face temperature (Figures 4&5a–c), salinity (Figures 4&5d-
f), chlorophyll (Figures 4&5g-i) and nitrate (Figures 4&5j–l)
in our 3 regions of interest. The ESMs reasonably reproduce
the annual cycle of surface temperature, but the annual cycles
of salinity, chlorophyll and nitrate are not simulated well in
any of them (see supporting Figures S1- S5) and the range of
simulated salinity and biological properties is large.

Temperature is relatively consistent between model ensem-
bles (Figure 4a–c), but with large variability between models
(Figure 5a-c). An annual, positive bias occurs in the GoM
(bias = +2.30°C, Figure 4a), whereas temperatures are over-
estimated in winter (Dec–Feb) on the SS and GB (bias =
+1.95 and +0.94°C respectively, Figure 4a-c) and underes-
timated in summer (Jun–Aug) on GB (-1.53°C, Figure 4f).
The range of simulated surface salinity is large (Figure 4d–f).
Most models overestimate salinity in the GoM (bias = +1.46,
Figure 4d). The mismatch is large on the SS and GB but not
consistent among models, except for an annual, positive bias
in CMIP6 models (bias = +1.42 and +0.76 respectively, Fig-
ure 4e–f). In the two latter regions, the biases in CMIP5 mod-
els compensate each other, resulting in an ensemble mean
close to the observations.

For surface chlorophyll, there is a large discrepancy between
the model ensembles and observations (Figure 4g–i). Inter-
model differences are largest for the time of maxima and
magnitude of the spring and fall blooms (Figure 5g–i, sup-

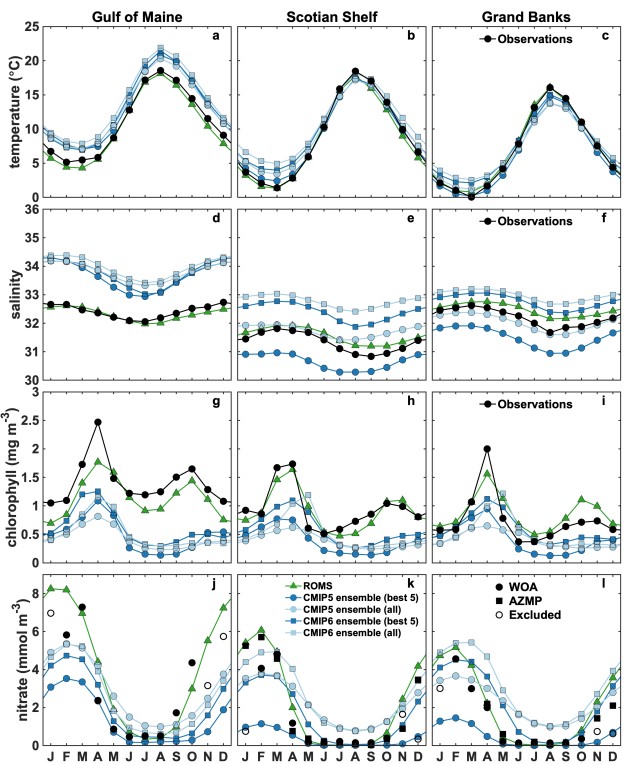

**Figure 4.** Observed, ROMS and ensemble means area averaged sur-
face temperature (a-c), salinity (d-f), chlorophyll (g-i) and nitrate
(j-l) in the 3 NWA shelf regions. Nov–Jan WOA nitrate data are ex-
cluded (open circles). Model comparison with observations in the
Gulf of Maine is therefore only available from February to Octo-
ber. For the Scotian Shelf and Grand Banks additional AZMP data
are available. In case of multiple observations, the data are monthly
averaged.

porting Figures S1–S5g–i). Standard deviations for the mag-
nitude of the spring bloom are large among ESMs in the 3
zones (SD=0.6, 0.81 and 0.83 mg m$^{-3}$ in GoM, SS and GB,
respectively). The maxima of the spring bloom also vary sig-
nificantly in time among the models, with a standard devia-
tion among ESMs for the time of maxima of the bloom of
about 1.5 months (SD=1.15, 1.59 and 1.62 months in GoM,
SS and GB, respectively). Most models in the CMIP5 group
do not simulate a fall bloom, hence none is present in the
ESM ensemble mean, but rather a fall/winter increase in
chlorophyll concentrations. Among the CMIP6 group, only
models 23–25 generate a fall bloom (see supporting Figures
S4–S5 g–i). Overall, the ESMs underestimate annual surface
chlorophyll concentrations (bias = –0.94, –0.50 and –0.29
mg m$^{-3}$ for GoM, SS and GB, respectively, Figure 4g–i ). The
chlorophyll bias is about 20% smaller in the CMIP6 group
compared to CMIP5.

There are also large discrepancies between the model en-
sembles and observations for nitrate (Figure 4j–l), particu-
larly in the CMIP5 group. The variability in nitrate concen-

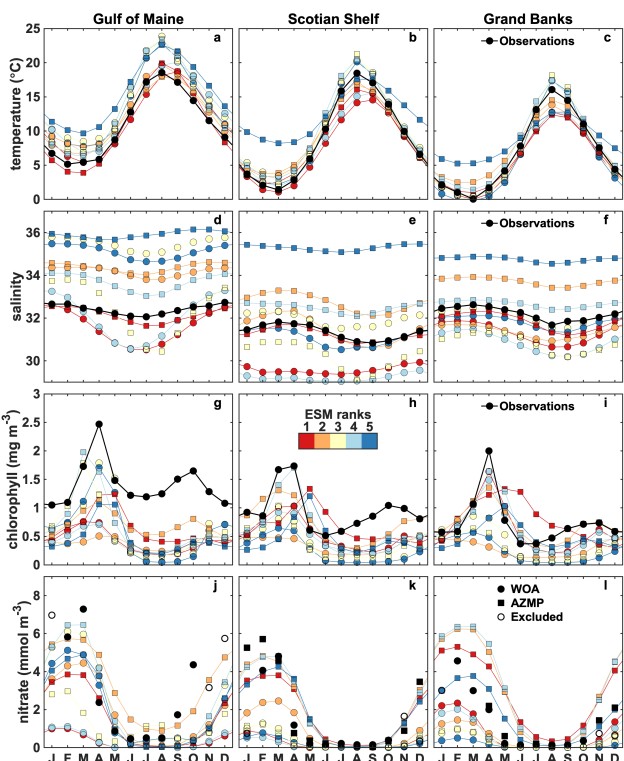

**Figure 5.** Observed (black dots) and best ESMs area averaged surface temperature (a-c), salinity (d-f), chlorophyll (g-i) and nitrate (j-l) in the 3 NWA shelf regions. The colored circles and squares indicate the CMIP5 and CMIP6 models, respectively. Nov–Jan WOA nitrate data are excluded (open circles). Model comparison with observations in the Gulf of Maine is therefore only available from February to October. For the Scotian Shelf and Grand Banks additional AZMP data are available. In case of multiple observations, the data are monthly averaged.

trations among the ESMs is also large (SD = 2.80 mmol m$^{-3}$) but smaller by 29% in the CMIP6 group. Most of the models reproduce the seasonal variability of surface nitrate (Figure 5j–l, supporting Figures S1–S5j–l); however, the CMIP5 models tend to underestimate fall-winter concentrations (winter bias = −1.28 mmol m$^{-3}$), whereas the CMIP6 model group performs better but with some mismatches in the timing of the seasonal changes (spring, fall). Note that since Nov.–Jan. nitrate WOA observations were excluded from the analysis (see section 2.1.5), winter observations are only available in February in the Gulf of Maine and in December and January in Grand Banks. A few models markedly overestimate surface nitrate concentrations in the NWA shelf regions (see supporting Figures S1, S3–5), including within the CMIP6 group. Supporting Figures S6–S9 provide an illustration of the model variability for chlorophyll and nitrate in March (Figures S6 and S7) and October (Figures S8 and S9), i.e. around the time of the spring and fall blooms respectively.

The regional ACM well reproduces the annual cycle of surface temperature (Figure 4a–c), salinity (Figure 4d–f), chlorophyll (Figure 4g–i) and nitrate (Figure 4j–l) in the three regions. The model correctly simulates the overall magnitude of temperature and chlorophyll biomass, the timing of the maxima of spring and fall blooms and the latitudinal variations in temperature, salinity, chlorophyll and nitrate, although the magnitude of the spring bloom in the GoM and GB regions is underestimated. Late summer surface salinity is slightly overestimated on the SS and GB.

## 3.2 Model statistics

Error statistics, i.e. RMSD and bias, are now analyzed and used to calculate the model scores. The distribution and relationships between scores are explored and then the ranks calculated.

Except for the relationship between temperature and salinity RMSD (r = 0.82, $p < 0.001$), the RMSD between the spatially averaged climatological observations and models are not consistent between variables, as indicated by the increasing temperature RMSD in Figure 6. However, temperature and chlorophyll RMSD are weakly correlated (r = 0.50, p = 0.005). For temperature and salinity, models 3, 20–21, and 24–25 have the largest discrepancy with observations and some clearly represent better the annual cycle than others. The best models for temperature (5–6, 14, 16 28) do not always match the best for salinity (5, 16, 27–28, 30). For chlorophyll, the largest discrepancies with observations are in models 4, 8 14 and 19–21, but overall chlorophyll RMSD are relatively large and homogeneous, except for a few models that have lower RMSD (e.g. models 22–23). Interestingly, the magnitude of the spring bloom in model 18 (CMIP5 group) is somewhat close to the observations. However, the time shift of the bloom (May–June) results in a poor agreement with observations. The mismatch between observed and simulated nitrate is much higher for models 5, 7, 18 and 29 and some models are much better at representing the observed annual cycle (Figure 6), as indicated by the lower RMSD. The RMSDs of the ACM are about a third of the average RMSD of the ESMs for both chlorophyll (ESM RMSDs are ×2.0–4.1 that of the ACM) and nitrate (×1.4–11.4), a quarter for temperature (×1.1–10.4) and 13% for salinity (×1.3–15.5).

Model scores (see Sect. 2.3) represent the spatial and temporal mismatch within the NWA shelf region (Figure 7). In general, the scores provide similar results as the RMSDs in Figure 6, although groups tend to emerge from the score calculation. As observed previously in Figure 6, the scores of ESMs have a much larger range of variability for temperature (1.5–7.8), salinity (0.5–4.2) and nitrate (1.4–13.2) than for chlorophyll (0.81–1.42) due to the large mismatch observed with a few models (Figure 7, supporting Figures S1–S5). For temperature, 4 of the 6 poorest (largest) scores (> 4.5) are in the CMIP6 group. They all markedly overestimate tem-

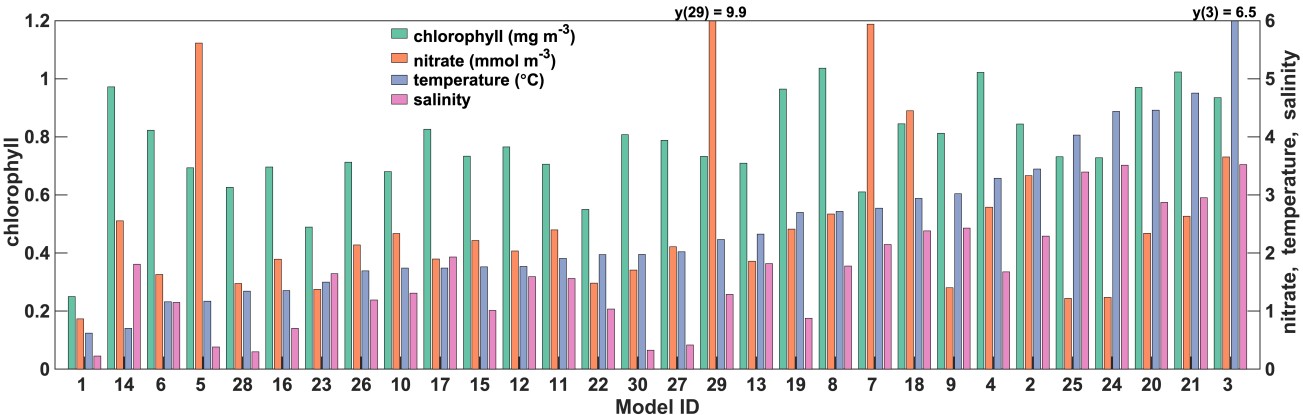

**Figure 6.** Root mean square difference between monthly, regionally averaged observations and models. Model numbers refer to the IDs in Table 1.

perature, especially in the GoM (see supporting Figures S1, S4–S5), except for model 4 that underestimates temperature in SS and GB. The other models have also the poorest scores with respect to salinity. They all largely overestimate salinity in the 3 regions and are clearly outliers with respect to their CMIP category. The range of variability in chlorophyll scores did not reduce from CMIP5 to CMIP6 and given the relatively low scores of a few CMIP6 models (i.e. 22 and 23), the range is larger in the CMIP6 group (0.8–1.4, Figure 7, right panel) than in the CMIP5 group (1–1.4, Figure 7, left panel). With the exception of model 29, which has a very poor (high) score for nitrate, the range of variability in nitrate is reduced in the CMIP6 group. In total, 5 models (3, 5, 7, 18, 29) have very poor scores for nitrate ($> 4$) strongly overestimating surface nitrate, except for model 3 in the Gulf of Maine (see supporting Figure S1j-l). The remaining models have more homogeneous nitrate scores (Figure 7) with the best (lowest) scores in models 25, 24, 9 and 6 (Table 2). Models that underestimate nitrate (2, 8, 14 and 19, see supporting Figures S1–S4) have a better score because they match the low nitrate observations in late spring–summer (Table 2). Overall, ACM has the best scores, $\overline{S}(\upsilon)$, for temperature (1.14), salinity (0.48), chlorophyll (0.64) and nitrate (1.27).

Among the 4 variables, and including the regional model, we found a correlation between the scores of temperature and salinity ($r = 0.74$, $p < 0.001$), as well as weak correlations between chlorophyll and temperature ($r = 0.53$, $p = 0.0025$) or salinity ($r = 0.42$, $p = 0.02$). There was no correlations between nitrate and chlorophyll (r = 0.03, p = 0.86$r = 0.53$, $p = 0.0025$), and nitrate and temperature ($r = 0.05$, $p = 0.78$) or salinity ($r = 0.003$, $p = 0.99$). As can be seen in Figure 6, the ESMs with a poor representation of nitrate are not necessarily performing poorly with respect to the other variables. Model 7 for instance has the poorest score for nitrate and a relatively poor score for temperature and salinity but the best score of the CMIP5 group for chlorophyll (Figure 7, left panel). Model 5 has a poor score for nitrate but

among the best scores for temperature and salinity. In fact, only models 3 and 18 have poor scores for all variables. Similarly, models 24 and 25 have the best scores for chlorophyll but are among the worst for temperature and salinity. On average, models have worse scores in the GoM (3.99, 2.49, 1.73, 3.15) than on the SS (3.36, 2.35, 0.94, 2.22) and GB (2.53, 1.41, 0.72, 2.47) for temperature, salinity, chlorophyll and nitrate, respectively.

Overall, 4 groups emerge on the chlorophyll-nitrate space in Figure 7. This grouping is somewhat arbitrary but provides a "biological" focus on model performance that can be related to the biological ranking ($\overline{R}_{bio}$) in Table 2. It also follows the general ranking presented in Figure 8, with a few exceptions. Group A includes 11 of the 14 best models (5 CMIP5 and 6 CMIP6) except for model 9 and 24–25 whose rankings are degraded due to poor representation of temperature and salinity. Within the 14 best models, the 3 models that are not included in Group A are model 5 and 15–16, which have mid to poor nitrate scores but are among the best models for temperature and salinity. Group B includes 4 intermediate-score models with respect to biology (15, 16, 17, 2). Group C includes the 8 models with poor chlorophyll scores (5 CMIP5 and 3 CMIP6) and Group D the 5 models with poor nitrate scores (4 CMIP5 and 1 CMIP6). Most of the models with poor scores for temperature and/or salinity are included in Group C, i.e. with the poor chlorophyll scores.

The overall model ranking (average of temperature, salinity, chlorophyll and nitrate ranks) indicates the gap between ACM and ESMs, as well as within ESMs (Figure 8). As expected, ACM ranks first, following the best scores for both chlorophyll and nitrate. The gap between ACM and model 28 (ESM with best $\overline{R}$ and $\overline{R}_{bio}$, Table 2) indicates that none of the ESM performs best for all fields, especially for both chlorophyll and nitrate. This is also shown by the large range in individual ranks (dark grey lines in Figure 8) in most models. Group A includes the 8 best ranking models, 2 from CMIP5 (6, 10) and 6 from CMIP6 (28, 23, 22, 26, 27, 30,

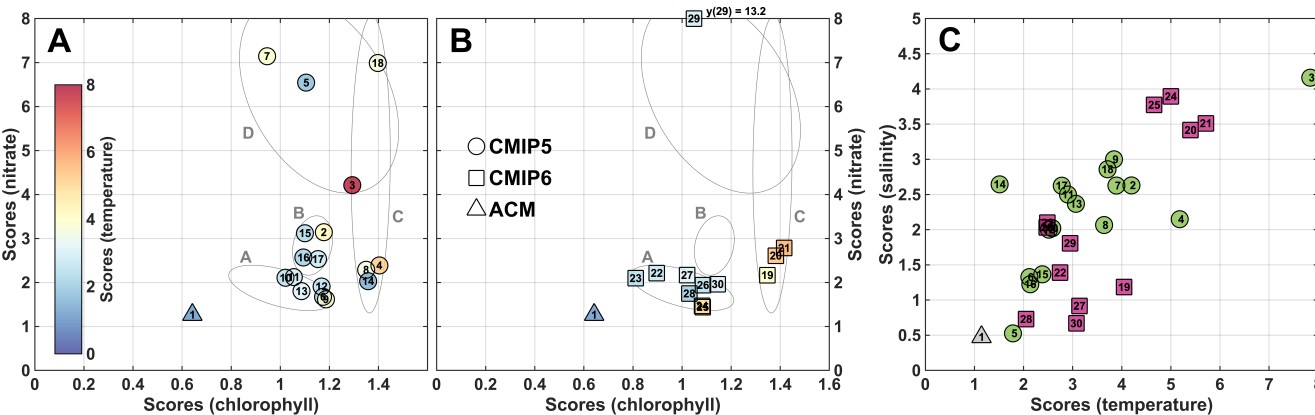

**Figure 7.** Model scores for surface chlorophyll (x-axis), nitrate (y-axis) and temperature (color scale) for the CMIP5 group (A, left panel), the CMIP6 group (B, middle panel) and the regional model (triangles). Salinity and temperature scores are compared in panel C (right). The grey ellipsoids indicate the groups A–D (see text) and are the same in panels A and B.

respectively). The most consistent in term of individual and overall ranking is model 28 (best ESM), the other ones having a relatively large spread. On the other side of the spectrum, models 18, 20, 3 and 21 (Groups C and D) have the poorest ranks because of their consistently poor scores. Model 2 has also consistent poor ranks for all variables. Despite its poor performance with respect to nitrate, model 29 is ranked within the mid-range of the ESMs because of the better performance with respect to the other variables (ranks 8–15); model 7 has consistently poor performances except for chlorophyll (rank 4).

Model scores and ranking were also calculated along the boundaries of the regional model (see supporting Figure S10). The ranking shows that model performance on the shelf is not necessarily indicative of the performance along the boundaries of the regional model (supporting Figure S11, Table S2). Moreover, individual rankings are much more variable at the boundaries, even for the best performing models. The 8 best ESMs along the boundaries (22, 11, 30, 28, 16, 10, 26, 6) have an average rank of 9.2–10.5. There are no significant correlations between individual rankings, including temperature and salinity. Nonetheless, there is some agreement between the shelf and the outer boundary ranking for chlorophyll ($\rho = 0.80$), nitrate ($\rho = 0.81$) and salinity ($\rho = 0.81$, supporting Figure S12 and Table S3). Interestingly, the agreement is better with CMIP6 models (Table S3). However, there is no agreement for temperature. A similar pattern is found for individual boundaries (Figure S13). In this case, and apart from temperature, the model ranks along the northeastern boundary agree the most with those from the shelf.

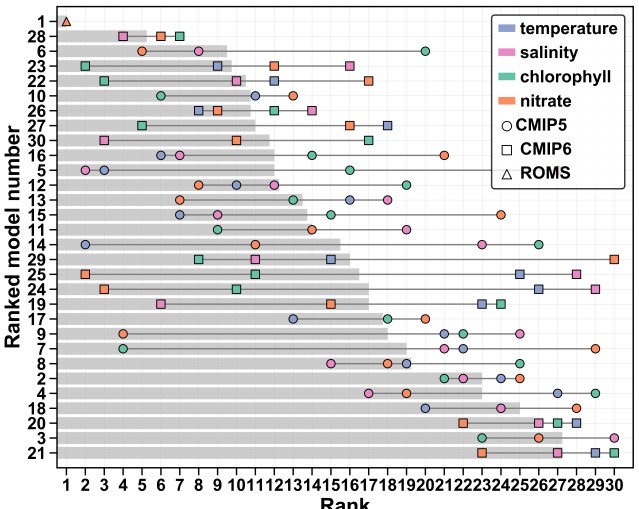

**Figure 8.** Model average (grey bars) and specific (dots) ranking. The final ranking is shown on the y-axis. Hidden coinciding ranks (models 2, 3, 6, 10, 11, 18, 27, 28 and 30) are provided in Table 2.

### 3.3 Additional model-data comparisons for regional ACM

While the resolution of the ESMs does not allow for a comparison at smaller spatial scales, we further compare the regional ACM to cross-shelf transects and station observations (Figure 9) along the Halifax Line (see Figure 1). The ACM reproduces the seasonal variation and the vertical gradient in chlorophyll and nitrate along the transect (Figure 9), although the simulated distributions are smoother than the glider observations. The summer subsurface chlorophyll maximum is located at the appropriate depth (28 m simulated versus 32 m observed, on average). The ACM somewhat underestimates the depth of the nitracline in the offshore waters

**Table 2.** Annual model scores and ranking. $\overline{R}$ represents the multi variable mean ranking and $\overline{R}_{bio}$ the chlorophyll and nitrate mean ranking. The final rank is provided in the right column. The asterisks beside the overall rank indicate a possible overestimation of the rank due to low nitrate concentrations (Figures S1–S5j–l).

| Ranked models | | | Scores | | | | Ranks | | | | | | |
|---|---|---|---|---|---|---|---|---|---|---|---|---|---|
| Name | ID | CMIP | Temp. | Salt. | Chl-a | NO3 | Temp. | Salt. | Chl-a | NO3 | $\overline{R}$ | $\overline{R}_{bio}$ | Overall |
| ACM | 1 | – | 1.14 | 0.48 | 0.64 | 1.27 | 1 | 1 | 1 | 1 | 1.0 | 1.0 | 1 |
| MPI-ESM1-2-HR | 28 | 6 | 2.05 | 0.73 | 1.03 | 1.75 | 4 | 4 | 7 | 6 | 5.3 | 6.5 | 2 |
| GFDL-ESM2G | 6 | 5 | 2.12 | 1.33 | 1.17 | 1.67 | 5 | 8 | 20 | 5 | 9.5 | 12.5 | 3 |
| GFDL-ESM4 | 23 | 6 | 2.49 | 2.10 | 0.81 | 2.10 | 9 | 16 | 2 | 12 | 9.8 | 7.0 | 4 |
| CNRM-ESM2-1 | 22 | 6 | 2.74 | 1.39 | 0.90 | 2.21 | 12 | 10 | 3 | 17 | 10.5 | 10.0 | 5 |
| HadGEM2-CC | 10 | 5 | 2.58 | 2.02 | 1.02 | 2.11 | 11 | 13 | 6 | 13 | 10.8 | 9.5 | 6 |
| IPSL-CM6A-LR | 26 | 6 | 2.47 | 2.03 | 1.09 | 1.94 | 8 | 14 | 12 | 9 | 10.8 | 10.5 | 7* |
| MIROC-ES2L | 27 | 6 | 3.14 | 0.92 | 1.02 | 2.17 | 18 | 5 | 5 | 16 | 11.0 | 10.5 | 8* |
| UKESM1-0-LL | 30 | 6 | 3.08 | 0.67 | 1.15 | 1.96 | 17 | 3 | 17 | 10 | 11.8 | 13.5 | 9 |
| CNRM-CM5 | 5 | 5 | 1.78 | 0.53 | 1.11 | 6.54 | 3 | 2 | 16 | 27 | 12.0 | 21.5 | 10 |
| MPI-ESM-MR | 16 | 5 | 2.14 | 1.22 | 1.09 | 2.57 | 6 | 7 | 14 | 21 | 12.0 | 17.5 | 11 |
| IPSL-CM5A-LR | 12 | 5 | 2.52 | 2.00 | 1.17 | 1.91 | 10 | 12 | 19 | 8 | 12.3 | 13.5 | 12 |
| IPSL-CM5A-MR | 13 | 5 | 3.07 | 2.37 | 1.09 | 1.80 | 16 | 18 | 13 | 7 | 13.5 | 10.0 | 13 |
| MPI-ESM-LR | 15 | 5 | 2.38 | 1.37 | 1.10 | 3.12 | 7 | 9 | 15 | 24 | 13.8 | 19.5 | 14 |
| HadGEM2-ES | 11 | 5 | 2.90 | 2.50 | 1.06 | 2.12 | 14 | 19 | 9 | 14 | 14.0 | 11.5 | 15 |
| IPSL-CM5B-LR | 14 | 5 | 1.51 | 2.64 | 1.36 | 2.03 | 2 | 23 | 26 | 11 | 15.5 | 18.5 | 16* |
| NorESM2-LM | 29 | 6 | 2.95 | 1.81 | 1.05 | 13.23 | 15 | 11 | 8 | 30 | 16.0 | 19.0 | 17 |
| GISS-E2-1-G-CC | 25 | 6 | 4.66 | 3.77 | 1.08 | 1.44 | 25 | 28 | 11 | 2 | 16.5 | 6.5 | 18 |
| CanESM5 | 19 | 6 | 4.05 | 1.18 | 1.35 | 2.16 | 23 | 6 | 24 | 15 | 17.0 | 19.5 | 19* |
| GISS-E2-1-G | 24 | 6 | 5.00 | 3.89 | 1.08 | 1.47 | 26 | 29 | 10 | 3 | 17.0 | 6.5 | 20 |
| MRI-ESM1 | 17 | 5 | 2.78 | 2.63 | 1.15 | 2.53 | 13 | 20 | 18 | 20 | 17.8 | 19.5 | 21 |
| GISS-E2-R-CC | 9 | 5 | 3.84 | 3.00 | 1.19 | 1.62 | 21 | 25 | 22 | 4 | 18.0 | 13.0 | 22 |
| GFDL-ESM2M | 7 | 5 | 3.89 | 2.63 | 0.95 | 7.14 | 22 | 21 | 4 | 29 | 19.0 | 16.5 | 23 |
| GISS-E2-H-CC | 8 | 5 | 3.64 | 2.07 | 1.35 | 2.29 | 19 | 15 | 25 | 18 | 19.3 | 21.5 | 24* |
| CanESM2 | 2 | 5 | 4.20 | 2.63 | 1.18 | 3.14 | 24 | 22 | 21 | 25 | 23.0 | 23.0 | 25 |
| CMCC-CESM | 4 | 5 | 5.18 | 2.15 | 1.40 | 2.39 | 27 | 17 | 29 | 19 | 23.0 | 24.0 | 26* |
| NorESM1-ME | 18 | 5 | 3.71 | 2.86 | 1.40 | 6.99 | 20 | 24 | 28 | 28 | 25.0 | 28.0 | 27 |
| CESM2 | 20 | 6 | 5.40 | 3.42 | 1.38 | 2.61 | 28 | 26 | 27 | 22 | 25.8 | 24.5 | 28 |
| CESM1-BGC | 3 | 5 | 7.84 | 4.16 | 1.29 | 4.21 | 30 | 30 | 23 | 26 | 27.3 | 24.5 | 29 |
| CESM2-WACCM | 21 | 6 | 5.71 | 3.51 | 1.42 | 2.78 | 29 | 27 | 30 | 23 | 27.3 | 26.5 | 30 |

(34 m versus 43 m, $x > 150$ km) and overestimates surface nitrate in spring and fall, as seen in Figure 4.

Station 2, which is located nearshore on the Halifax Line (see Figure 1), provides additional, vertically resolved information with high temporal resolution that is useful for model validation (Figure 10). At this location, the ACM reproduces the annual cycle of chlorophyll and nitrate. Surface and subsurface nitrate and chlorophyll are qualitatively reproduced in all seasons except during the spring bloom, which is more pronounced and reaches deeper in the observations, although the magnitude and vertical distribution of chlorophyll concentration agree well with the glider observations at this time. A quantitative, point-to-point comparison of the ACM with the time series and glider observations along the Halifax Line (Figure 9) and at Station 2 (Figure 10) is provided in Table 3. The comparison indicates relatively high correlations be-

tween the ACM and time series of chlorophyll (0.68–0.78) and nitrate (0.83–0.92) along the Halifax Line as well as glider measurements of chlorophyll (0.85–0.94) for all seasons. Correlations are high as well at Station 2 for nitrate time series and glider measurements of chlorophyll. The largest discrepancies with observations are found with the time series of chlorophyll in spring. These results indicate an overall good skill of the model to reproduce the seasonal, vertically resolved observations on the Scotian Shelf.

## 4 Discussion

### 4.1 Overall model performance on the shelf

There are significant discrepancies with observations and a large variability among ESMs in the representation of surface

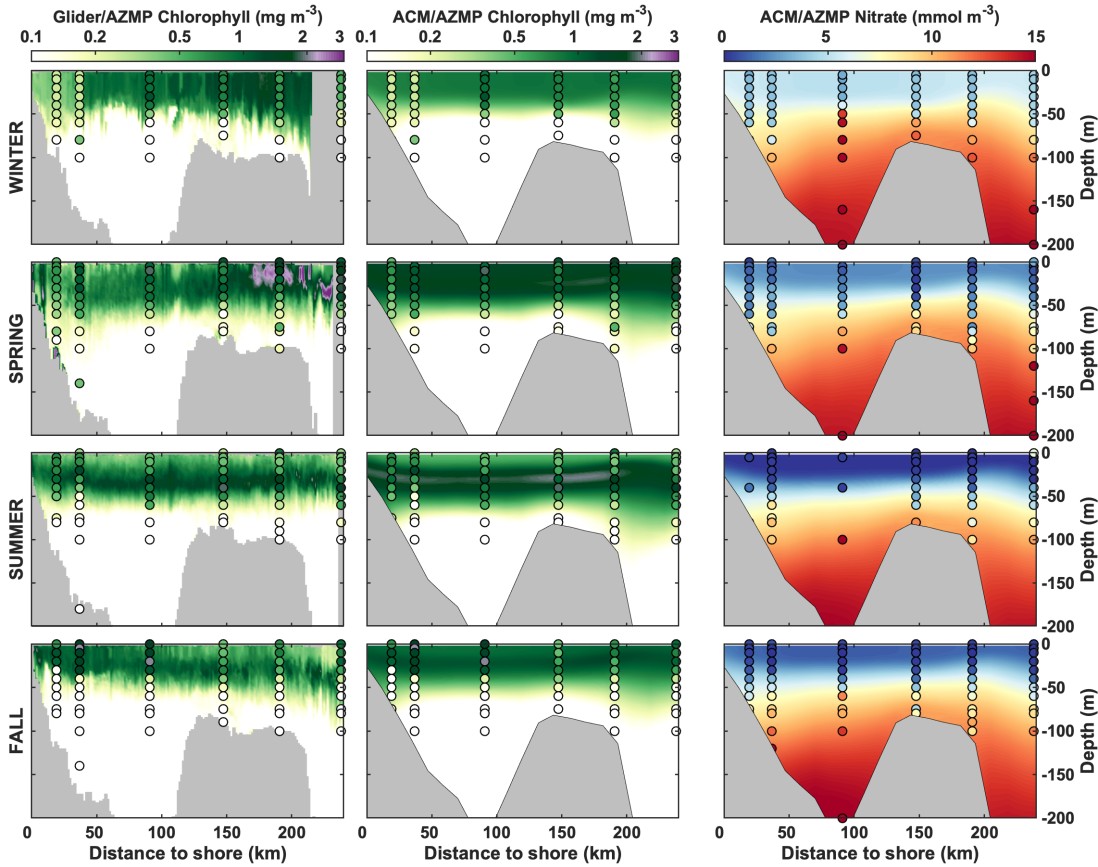

**Figure 9.** Comparison of gliders, AZMP and model seasonal climatologies of chlorophyll and nitrate along the Halifax line.

**Table 3.** Comparison statistics between ACM and AZMP and glider observations along the Halifax Line and at Station 2.

| Season[*] | RMSD | | | | Bias | | | | Correlation coefficient | | | |
|---|---|---|---|---|---|---|---|---|---|---|---|---|
| | W | S | S | F | W | S | S | F | W | S | S | F |
| | | | | | | | Halifax Line | | | | | |
| Chlorophyll (time series) | 0.25 | 0.37 | 0.39 | 0.36 | 0.08 | 0.22 | 0.28 | 0.13 | 0.68 | 0.78 | 0.71 | 0.75 |
| Chlorophyll (Glider) | 0.22 | 0.42 | 0.25 | 0.22 | -0.14 | 0.13 | 0.17 | 0.04 | 0.88 | 0.78 | 0.94 | 0.85 |
| Nitrate | 2.99 | 2.73 | 2.13 | 1.77 | 0.76 | 2.03 | 0.74 | 1.27 | 0.90 | 0.83 | 0.85 | 0.92 |
| | | | | | | | Station 2 | | | | | |
| Chlorophyll (time series) | 0.26 | 1.74 | 0.52 | 0.30 | 0.05 | -0.56 | 0.26 | 0.01 | 0.64 | 0.22 | 0.48 | 0.82 |
| Chlorophyll (Glider) | 0.15 | 1.06 | 0.31 | 0.17 | -0.03 | -0.46 | 0.25 | 0.02 | 0.87 | 0.69 | 0.91 | 0.93 |
| Nitrate | 0.96 | 1.57 | 1.58 | 1.37 | 1.19 | 1.62 | 0.26 | 0.58 | 0.85 | 0.86 | 0.91 | 0.94 |

[*]Seasons are order sequentially and abbreviated as W (winter, Dec–Feb), S (spring, Mar–May), S (summer, Jun–Aug) and F (fall, Sep–Nov).

temperature, salinity, chlorophyll and nitrate in the NWA shelf (Table 2, Figure 6 and supporting Figures S1–S5). A warm bias and a general overestimation of surface salinity in most models indicate a mismatch in the location of the Gulf Stream that influences conditions on the shelf, in line with the previous results of Loder et al. (2015) and Saba et al. (2016). Chlorophyll concentration was also systematically underestimated, whereas surface nitrate concentration is relatively

variable between models. These patterns agree with the qualitative assessment of Lavoie et al. (2013, 2019). The spring and fall blooms, which are characteristic annual features of the NWA region (Greenan et al., 2004, 2008) are absent in some and most models, respectively. The correlation between temperature and chlorophyll scores (and to a lesser extent salinity) and the concomitant poor scores in chlorophyll and temperature/salinity (i.e. Group C in Figure 7) indicate that

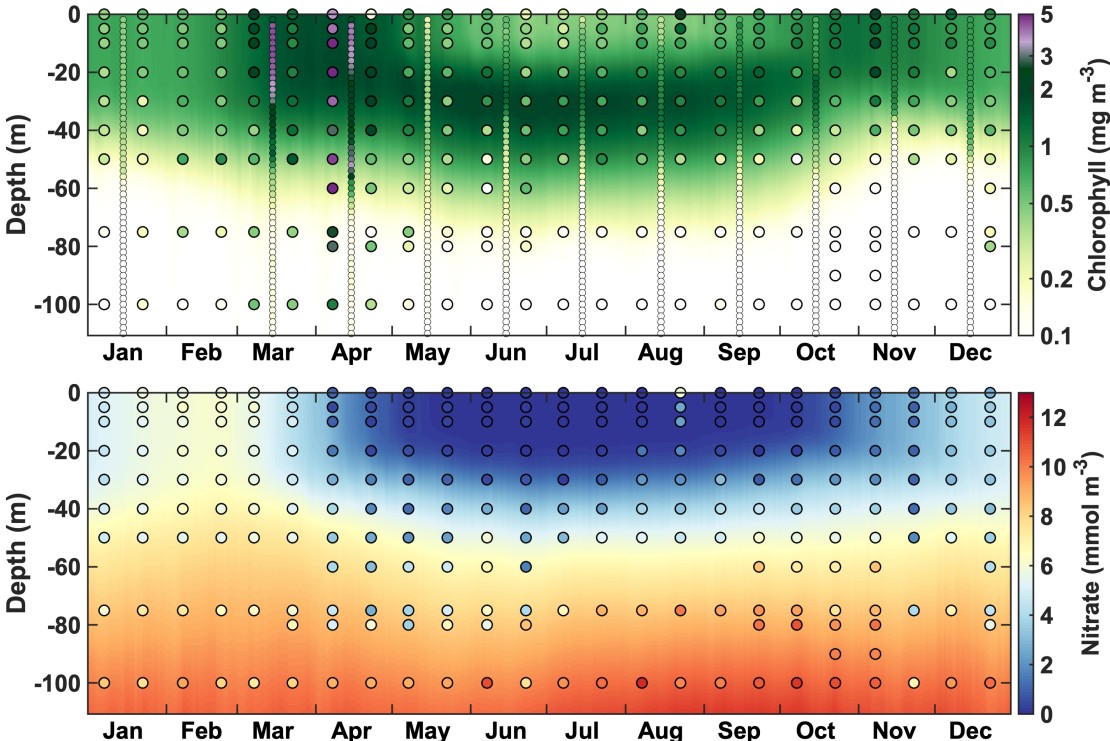

**Figure 10.** Comparison of vertically-resolved time series of chlorophyll (top) and nitrate (bottom) at Station 2 from the regional model (background), the glider transects (small dots) and the bimonthly sampling (large dots).

errors in surface chlorophyll concentration are partly driven by a misrepresentation of the general circulation and, more generally, of ocean physics. The improvement in chlorophyll from CMIP5 to CMIP6 in some models without an associated improvement in temperature (see below) suggest that the errors in surface chlorophyll were also driven to some extent by errors in the biogeochemical model component. Lavoie et al. (2019) indicated that the misrepresentation of primary production in the NWA may be associated with the misrepresentation of particulate organic matter sinking and remineralization in the subsurface layer. They found an annual subsurface nitrate peak in CanESM2, GFDL-ESM2M, NorESM1-ME, CESM1-BGC (models 2, 7, 18 and 3, respectively) similar to the high surface nitrate found in this study (supporting Figures S1 and S3). However, all these models had poor scores in our assessment and therefore do not provide an appropriate representation of the biogeochemistry on the NWA shelf (Figure 8) or along the ACM boundaries (Figure S11). However, it is not possible, and beyond the scope of this work, for us to draw conclusions about the source of the regional mismatch in surface chlorophyll and nitrate in the ESMs.

Following Rickard et al. (2016), who used a similar ranking procedure, the 29 ESMs can be divided into an inner and an outer model ensemble of the NWA shelf. The outer ensemble includes 18 models that clearly misrepresent surface condi-

tions in the NWA shelf (models 2–5, 7–9, 11, 14–15, 17–21, 24–25 and 29) and were selected as follows. The 7 models with lowest ranks (2–4, 8, 18, 20–21) were included because they consistently misrepresent all surface fields on the NWA shelf. Models 7, 9, 17 had poor scores for three variables, Model 15 was also included in the outer ensemble because of the misrepresentation of surface nitrate, whereas models 24–25 misrepresented temperature and salinity. Since nitrate scores neither correlate with chlorophyll nor temperature, the mismatch with nitrate observations is more likely related to intrinsic biogeochemical model behaviour rather than to a mismatch in circulation, as suggested by Lavoie et al. (2019). Models with persistent positive or negative biases in surface nitrate (4–5, 7–8, 11, 14, 19 and 29, Figures S1–S5) were selected because they misrepresent the seasonal nitrate dynamics and therefore the other biogeochemical variables driven by nitrate are questionable. Seven of the outer models were different generations (CMIP5 and CMIP6) of the same model, i.e. CanESM (2, 19), CESM (3, 20–21) and NorESM (18, 29), which had also low ranks along the ACM boundaries. Their large scores imply that they have fundamental issues with representing biogeochemistry in the NWA.

The inner ensemble includes 11 models (6, 10, 12–13, 16, 22–23, 26–28, 30, Table 1). Can those be used as a multi-model (optimal) ensemble to characterize the future state of the NWA shelf region? Unfortunately, we found that an en-

semble mean of the best CMIP5 or CMIP6 models poorly represent historical surface fields due to the large variability within the ensemble (Figure 5) and the biases in the ensemble surface temperature, salinity and chlorophyll concentration (Figure 4). Model 28 (MPI-ESM1-2-HR, CMIP6) was the only ESM with good performances for all variables and is therefore the most appropriate to represent surface conditions in the NWA shelf.

The regional model clearly outperformed the ESMs in our assessment, with a consistent representation of the surface and subsurface fields in all shelf areas. The high spatial resolution of the regional model also allowed for a fine scale model validation that was not possible for the ESMs. The complementary glider transects and time series stations provide a high-resolution dataset of in-situ chlorophyll and nitrate concentrations and shows that the regional model resolves seasonal and vertical variation in chlorophyll and nitrate on the Scotian Shelf, something that none of the ESMs were able to reproduce.

## 4.2    Model performance along the regional model boundaries

The assessment of an ESM's performance on the NWA shelf, as presented above, is necessary prior to using its results, for example, to estimate historical and future trends in physical and biogeochemical tracers (Lavoie et al., 2013, 2019) and their effects on upper trophic levels (e.g., Bryndum-Buchholz et al., 2020b; Stortini et al., 2015). For regional downscaling, an ESM's performance along the boundaries of the regional model are critical (e.g., Lavoie et al., 2020). We found significant differences between model performance on the shelf and along the ACM boundaries and more variability in model performance for the latter. At the boundaries, all models have at least 1 variable that is poorly represented (Figure S11). Surprisingly, there is no relationship between ESM ranking on the shelf and at the ACM boundaries for temperature. Given the importance of large-scale circulation in the region some agreement was expected. The mismatch could be explained by a lesser control of large-scale currents on shelf temperature, although ESM biases for temperature and salinity on the shelf indicate the influence of the Gulf Stream. The agreement is better for the other variables (Table S3). Among the 10 best ESMs along the ACM boundaries, 8 are included in the inner ensemble described above; the best overall ESM on the shelf (model 28) is ranked third at the boundaries. Similarly, models with poor performances on the shelf (3, 18, 20–21) had also poor scores at the boundaries. The inner ensemble can therefore be used as a guide for ESM selection in the NWA region.

## 4.3    Uncertainties in score calculations

We used a heterogeneous dataset to calculate error statistics. Also, the regional model simulated the period 2000–2014, whereas the time range 1976–2005 was used with the CMIP models, for consistency in their comparison. For surface salinity, chlorophyll and nitrate, Lavoie et al. (2013) found negligible historical trends (1970s–2000s) in a multi-model comparison. For surface temperature, they found an increase in temperature <0.5°C over this period, which is very small in comparison to the inter-model differences (Figures S1–5a–c). Also, surface temperature is overestimated in the GoM, whereas the trend would result in an underestimate. Hence, the scores should not be affected by time differences between model and observation datasets.

Since the period 2000–2014 is available for the CMIP6 models, we calculated the scores over this period to be consistent with the regional model simulation and the chlorophyll and temperature observations. The 2000–2014 scores are in agreement with the 1976–2005 scores described in section 3.2 (see supporting Figure S14), showing the robustness of our calculations despite the heterogeneous dataset. The only significant differences are with models 30 and 21, which have improved and degraded 2000–2014 scores for temperature, respectively. Model 21 remains at the last rank (Table 2) but the overall rank of model 30 (UKESM1-0-LL) could be somewhat higher than indicated in Figure 8.

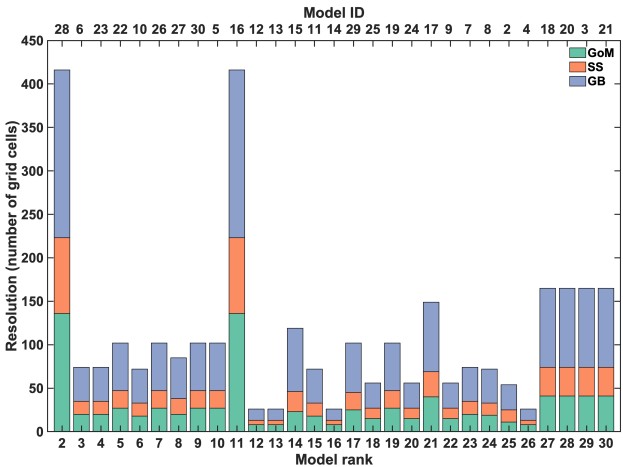

**Figure 11.** Resolution of the 29 ESMs ordered by their overall rank (see Figure 8).

## 4.4    Impact of spatial resolution

In general, the coarse horizontal resolution of the ESMs affects the representation of the NWA region in comparison to the regional model, particularly on the relatively narrow Scotian Shelf. The poor representation of coastal areas is a known limitation of global models (Holt et al., 2017) and results in a global underestimation of primary productivity in these regions (Bopp et al., 2013; Schneider et al., 2008).

There is no correlation between grid resolution and ESM rank (Figure 11) despite the fact that the best overall ESM

(MPI-ESM1-2-HR) has also the highest resolution (Table 1). This result shows that higher grid resolution, as called for by Lavoie et al. (2013) for the NWA and by McKiver et al. (2015) for the global ocean, is necessary but is not a guarantee for improved model performance at this time. In fact, some very coarse resolution models from the CMIP5 group were ranked as well or better than the other models and models with the second highest resolution (3, 18, 20–21) had all low ranks. The improved ranks at constant (e.g. models 22, 24, 25, 28) and even lower (model 29) ocean grid resolution in the CMIP6 group (Table 2, Figure 12) was also an indication that the discrepancies with observations, and the improvement in the CMIP6 models (see below), were not associated with the ocean grid resolution but rather resulted from the physical and biogeochemical setup of the models. Another hint at the lack of relationship between resolution and model rank is the limited improvement with the high-resolution MPI model in the CMIP5 group (MPI-ESM-MR), despite higher model grid resolution compared to its lower-resolution counterpart (MPI-ESM-LR, Table 2). The lack of correlation between model resolution and performance on the NWA shelf is not surprising as all ESMs are still coarse and do not explicitly resolve shelf-scale processes but rather rely on their parameterisation. Much higher resolution will be necessary to refine the projections in coastal areas (e.g., Holt et al., 2017; Saba et al., 2016), which is not currently computationally feasible in ESMs (Holt et al., 2009, 2017).

### 4.5   Impact of biogeochemical model structure

Although model performance is likely influenced by the biogeochemical model structure, we did not find a clear relationship between the type of biogeochemical model and performance. Here we only refer to the model type because the same model may have different parameterizations when used by different groups. While the inner and outer ensembles share only 3 biogeochemical models (PISCES, HAMOCC, TOPAZ2) out of 13, there was no indication of consistently better performance for the biogeochemical models in the inner ensemble. For example, models using similar ocean biogeochemistry (e.g., PISCES: 5, 12–14 (CMIP5), 22 and 26 (CMIP6), and HAMOCC: 15–16, 18 (CMIP5), 28–29 (CMIP6)) had very different ranks, with no obvious relationship between overall model rank and the ocean biogeochemical model component. Moreover, 5 and 4 biogeochemical models were represented in the 5 best ranked ESMs on the NWA shelf and outer ACM boundaries, respectively, similar to previous findings by Rickard et al. (2016). Lavoie et al. (2019) suggested that the PISCES biogeochemical model may underestimate subsurface remineralization in the CNRM and IPSL models, resulting in low surface nutrients where the Gulf Stream detaches from the coast. Our rankings (shelf and offshore) and the spatial patterns in Figures S1-9 do not fully support this hypothesis; high surface nitrate concentrations were present in the CNRM models throughout the region, whereas concentrations in the IPSL-CM5A models were low (except around the GoM in Spring) (Figures S1–4, S7, S9). It is unlikely that these large scale patterns are driven by upwelled Gulf Stream waters, although differences in remineralization could influence these general patterns.

### 4.6   Improvement from CMIP5 to CMIP6

Model performance improved in the new CMIP generation, but not uniformly across models and variables. We note that 2 of the 5 best models are from the CMIP5 for both the shelf and the ACM boundaries rankings. Therefore, with respect to historical conditions in the NWA region, CMIP6 models do not always have better performance. The average rank was not very different between the two CMIP groups, i.e. $\overline{R}$ = 16.8 and 14.9 for CMIP5 and CMIP6, respectively (Figure 8, Table 2). The change in performance between the two generations of models can be assessed by evaluating the subset of models that are available for CMIP5 and CMIP6. There are nine such models (Figure 12). All CMIP6 models have improved overall ranks, indicating better performance (Figure 12). The overall improvement was large only for models that had average to low ranks in the CMIP5 group (ranks 15–22, x-axis in Figure 12). Temperature and salinity did not improve except for GFDL-ESM2M and NorESM2-LM (and CanESM5 for salinity) and degraded in some cases. Models with poor scores for temperature and salinity (CESM2, GISS-E2-1-G-CC) had already poor scores in their CMIP5 version and therefore the cause of their poor performance is likely the same. The change in ranking is therefore mainly associated with better surface fields for chlorophyll and nitrate. This is particularly the case for model pairs 3, 5, 6 and 8, which ranked much better for chlorophyll (+8.2) and nitrate (+11.0) in the CMIP6 group (Figure 12). The chlorophyll rank in model pair 4 improved significantly (+18) but this improvement was counteracted by degraded temperature and nitrate ranks. The lack of general improvement in surface temperature indicates that the temperature bias detected in the CMIP5 group was not solved in CMIP6, as seen in Figure 4. We can only speculate about the source of improvement in the CMIP6 models. For specific changes in the CMIP6 model versions, the reader is referred to the references listed in Table 1. Kwiatkowski et al. (2020) recently showed that projected surface temperature, nitrate and net primary production differ significantly in CMIP5 and CMIP6 model ensembles. Higher climate sensitivity in CMIP6 models partly explains this difference but the source of change in primary production was not resolved. In the historical simulations, better surface chlorophyll and nitrate fields in CNRM-ESM2-1 may be associated with the transition from a climate model with ocean biogeochemistry to a fully coupled ESM, even though such transition may degrade historical simulations due to the replacement of observations by prognostic schemes that are poorly constrained (Séférian et al., 2019). Updated land and ocean biogeochemistry may

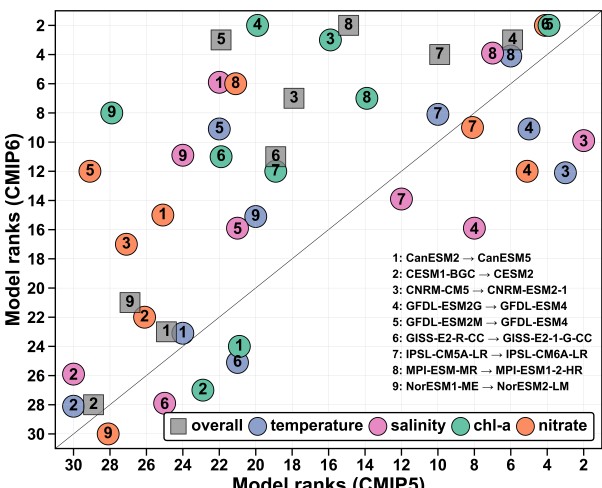

**Figure 12.** Comparison of the ranks of the former (x-axis) and current (y-axis) generations of ESMs. Grey squares represent the overall ranks, whereas the dots indicate temperature (blue), salinity (magenta), chlorophyll (green) and nitrate (orange) ranks. The numbers indicate the model (see legend). These numbers do not correspond to the original model IDs indicated in Table 1. The black line is the 1:1 line. Dots above this line indicate an improvement and dots below the line a worsening of the rank. Note that there were two CMIP5 GFDL models but only one in the CMIP6 group (model pairs 4 and 5).

have improved the representation of surface chlorophyll and nitrate in MPI-ESM1-2-HR (Müller et al., 2018), whereas the improvement in surface temperature and nitrate fields from GFDL-ESM2M to GFDL-ESM4 seem to be associated with the physical ocean component of the model, given that GFDL-ESM2G already performed well in the CMIP5 group. Danabasoglu et al. (2020) found a significant improvement for CESM2 at the global scale but a poor representation of the Gulf Stream–North Atlantic Current system, resulting in a large surface temperature bias. This is in line with our assessment for the NWA shelf where both physical and biological parameters had poor scores and the model was not found appropriate for shelf studies in the NWA.

### 4.7 Other coastal regions

Our results may also apply for other coastal regions, given the poor representation of coastal areas in ESMs, but the details are probably region specific. Discrepancies with observations in the NWA are partly driven by poor representation of large-scale circulation features such as the Gulf Stream and Labrador Current in most of the models. The representation of large-scale currents may improve (or worsen) in other regions, resulting in a different ranking there. For example, Rickard et al. (2016) found a different model selection in the inner model ensemble around New Zealand. Seven (out of 11) of their inner ensemble models (models 2–5, 7–8, 14)

are not included in our inner ensemble. Model 3, perhaps the best model in their assessment, ranked 29 out of 30 in the NWA shelf region (Figure 8, supporting Figure S1). The representation of the dynamic NWA circulation is a known issue in ESMs and further regional comparisons will be necessary to assess if our results are representative for the global coastal ocean.

## 5 Conclusions

We evaluated the CMIP5 and CMIP6 ESMs with biogeochemistry for the NWA shelf. Arguably, only 1 model (MPI-ESM1-2-HR) had a consistently good performance for all variables. 11 ESMs with satisfactory overall performance in their historical simulations of the NWA shelf were included in a ranked inner ensemble to guide the use of ESMs in the region. Apart for temperature, the ESMs evaluation along the boundaries of the regional model was relatively similar to the evaluation on the shelf but with more variability. Most of the highly ranked models can therefore be used either directly or for regional downscaling. We caution against using model ensembles that had poor agreements with observations on the NWA shelf. The regional model (ACM) clearly outperformed the global models and is a good candidate for downscaled projections in combination with one of the top ranked ESMs. Further refinement in the ACM should focus on the mechanisms that determine the magnitude of the spring bloom. Similar comparisons should be carried out in coastal areas before using CMIP model projections. While it is not clear how the presented model ranking will hold in other regions, it is highly likely that some models do not perform well in coastal areas generally and should not be used for regional investigations.

Given the lack of a direct relationship between model skill and horizontal resolution, it is unlikely that feasible grid refinement will significantly improve model performance in the NWA region. The improvement in scores from CMIP5 to CMIP6 shows that refining ocean biogeochemical components can improve the model performance.

*Code and data availability.* The ROMS code and the observations are available from the links referenced in the manuscript

*Supplement.* The supplement related to this article is available online at:

*Author contributions.* AL and KF conceived the study. AL and AK refined a previous version of the ACM model. AL conducted the analyses. AL wrote the manuscript with input from KF

*Competing interests.* The authors declare that they have no conflict of interest

*Acknowledgements.* The ACM was run on Compute Canada resources under the resource allocation project qqh-593-ac. We thank two anonymous reviewers for their constructive inputs that helped to improve the manuscript.

*Financial support.* We acknowledge funding from the Canada First Research Excellence Fund, through the Ocean Frontier Institute, the MEOPAR Network of Centres of Excellence through the Prediction Core, and an NSERC Discovery Grant held by KF.

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
