# Peer review of "An observation-based evaluation and ranking of historical Earth System Model simulations in the northwest North Atlantic Ocean"

_Biogeosciences, 2020_

## Referee Comment (RC1) · Anonymous Referee #1 · 6 Sep 2020

The study addresses the important question of whether the Earth system models can accurately simulate biogeochemical processes/fields on the Northwest North Atlantic shelf and if they are suitable for projecting the regional impact of climate change. The study ranks the CMIP5 and CMIP6 models in terms of capturing the observed seasonal variability in surface temperature, chlorophyll and nitrate in the Northwest North Atlantic shelf, highlighting that ESMs are not adequate to project accurately future changes in the NWA shelf. Based on the comparison with these ocean surface observations the study clearly demonstrates how a regional model outperforms the Earth system models

in terms of capturing the seasonal variability in the region during the historical period. The study then provides plausible reasons for the better performance of some of the Earth system models and the improvement from the CMIP5 to CMIP6.

The study is interesting, timely and I believe it is a substantial contribution towards improving our understanding and ability to provide accurate regional projections of climate change on the Northwest North Atlantic shelf, particularly in the context of biogeochemical fields. The methods and results are reasonable, justified and well presented. Hence, in my opinion the study fits well the Biogeosciences scope and should be consider for publication after addressing some very minor comments/revisions as I explain below.

**1   General comments (only one main query as I explain below):**

I agree that not all the ESMs are suitable to force regional model projections and should be selected carefully as not to force the regional model with unrealistic large-scale patterns.  The choice of the 'parent model' should be based on its performance on capturing the large scale patterns/features that are introduced in the regional models such as: large scale circulation patterns, and biogeochemical and physical fluxes in-out of the regional model domain (the study highlights well this point in the introduction, lines 48-59 and in discussion, lines 343-351). Misrepresentation of the ESMs surface fields in the Gulf of Maine, Scotian Shelf and Grand Banks may be due to the models not being able to accurately represent on-shelf fine-scale biogeochemical and physical processes due to lack of resolution (which is inadequate to resolve the physical/biogeochemical processes on the shelves in all the ESMs, as mentioned in the manuscript), lack of accurate river discharge, biogeochemical model parameter optimisation for global use rather than the region of interest, or lack/misrepresentation of other local processes (e.g., enhance mixing associated with tides). The regional model

is meant to address the above on-shelf problems, and as long as the large-scale patterns are introduced correctly at the open boundaries it should be able to perform well.

Hence, I am a little reserved about the conclusion that 3 specific ESMs are most appropriate to force regional model projections in NWA (lines 21-23 and 77-79) based on the ESMs' performance on the shelf at the surface. In my understanding, there is no explicit analysis in the study to link the ESMs ranking on the shelf with the large-scale physics/circulation patterns off-shelf, or across-shelf properties exchange beyond a moderate correlation of surface temperature and surface chlorophyll, and no explicit link between the ESMs performance on the shelf and their performance off the slope in the vicinity of the regional model boundaries. So in my opinion, I would suggest to further discuss and clarify this issue: does low ranking on the shelf translate directly to low ranking of the larger scale patterns in the region and specifically to low ranking in the quality of the ESMs exchange/input of physical and biogeochemical properties at the location of the open boundaries offshore the Scotian shelf, Gulf of Maine and Grand Banks? Why are the ESMs with high ranking on the shelf the ones that can provide the best open boundary conditions coming off the shelf?

**2  Specific comments (all minor):**

Line 75 (typo): I think an 'on' is missing such that: 'based on the mismatch . . .'

Lines 147-148 (just a suggestion): The authors can refer and link R and Rbio symbols in Table 2 directly with the rank for temperature, chlorophyll and nitrate, and the rank of chlorophyll and nitrate only, respectively, to help the readers follow more easily the ranking metric.

Line 179: Please can you clarify which months you consider for the winter bias for nitrate? In my understanding winter is defined as December-February (based on Table 3) but November-January are excluded from the observations for nitrate, so is it only

February?

Line 233 (just a suggestion): Maybe if you could clarify that here you mean the best overall ESM in terms of nitrate and chlorophyll (the Rbio in Table 2) by adding something along the lines: 'ACM and model 22 (the best overall ESM for combined nitrate and chlorophyll) indicates . . .'.

Lines 268-270: I agree that some of the errors in chlorophyll are linked with the temperature bias and the misrepresentation in general circulation. However, in my opinion i) the relative moderate correlation between chlorophyll and temperature r=0.51 and ii) the improvements in chlorophyll in CMIP6 relative to CMIP5 models that are not associated with any improvement in temperature (as discussed in lines 321-328) indicate that the errors in surface chlorophyll are also driven to a significant degree by a poor biogeochemical model component rather than by the ocean physics only. Hence, I suggest that the authors could modify this sentence to reflect this.

Section 4.2: Although the authors mention it, in my opinion, they could highlight even more that the lack of correlation between resolution and rank is not surprising as none of the ESMs have the resolution to explicitly resolve the processes in shelf-scales rather than parameterise them. Maybe the authors could add a sentence after line 303 along the lines: 'This lack of correlation between model resolution and accuracy on the NAW shelf and the primary control of performance by the model set-up is not surprising as all ESMs are coarse and do not explicitly resolve the shelf-scale processes but rather rely on their parameterisation.'

Figure 8 caption (just a suggestion): The authors mention that the temperature rank is hidden for model 6, however, in my understanding ranks for other models are also hidden as they coincide (model 1, 30 and 18). Maybe just for clarity and to make the link between figure 8 and table 2 explicit, you could mention in the caption something along the lines 'Coinciding ranks as shown in table 2 are hidden'.

---

## Referee Comment (RC2) · Anonymous Referee #2 · 14 Sep 2020

In this paper, the authors compare the output of CMIP5 and CMIP6 Earth System Models (ESMs) to observations in order to determine which models are suitable to build boundary conditions for projections. A ranking analysis was performed on a large array of ESMs. However, they are only looking at surface values of 3 variables and far away from the regional model boundaries, even though they mention on lines 44-46 that it is important to look at the information imposed at the boundaries. I think the objective stated on line 67 "Our objective is to assess the performance of a number of available ESMs in reproducing present conditions on the NWA shelf in contrast to a

high-resolution regional model" is more in line with what is presented in the manuscript since there is no analysis at the boundaries. They are not discussing the processes that lead to the observed values in the region under study and they are not analysing if the models do represent these processes correctly. I believe salinity should be included, as surface temperature depends strongly on atmospheric forcing while salinity is more representative of the different water masses in that region. Moreover, it is very surprising that a similar study (Lavoie et al. 2019) in the exact same area, with the same purpose, and using some of the same ESMs is hardly mentioned at all. No comparison of the results of this study with the 2019 study is made. Also there is not enough details on the comparison with the data, they appear to be comparing different time periods (see detailed comments) or on how the ESMs were brought to a single grid. There is only a vague mention of what the improvements are between the CMIP5 and CMIP6 models. What was improved should be stated (not only biogeochemistry of physics) so that the reader can judge on the potential impact on the ranking. Increasing the model resolution in order to improve the representation of the circulation in the NWA has been mentioned by many authors (e.g. Loder, Brickman, Yool). Here it is stated that the resolution does not have an impact. This is a big statement, considering the general agreement, and it should be demonstrated. The authors could show the changes in circulation of a few models they are giving in example for this. All these points should be addressed in order for the conclusions to be more convincing (ranking based on analysis of shelf surface conditions representative of boundary conditions). Also, Lavoie et al. (2019) estimated that the boundary conditions obtained with the ESMs were not as reliable for the simulation of the conditions on the Scotian Shelf and in the Gulf of Maine. It would good to know if there was an improvement in this regard with the CMIP6 ESMs. With regards to the level of information lacking in the manuscript I recommend a major revision before it can be published.

Abstract

Line 11: Here you say that the coarse resolution is not appropriate to represent the

circulation and elemental flux but later on you say that increasing the resolution does not matter. Is it important or not?

Line 14: ability to reproduce surface observations ...

Line 15: why is it particularly sensitive?

Line 16: The spatial mismatch in large-scale circulation was not demonstrated. There are references for CMIP5 but what about CMIP6. Changes, or not, in circulation should be shown/mentioned after an inspection of the ESMs results.

Line22: How can we say just by looking at the surface temperature, nitrate and chl a that the top three models are appropriate for boundary forcing? The model boundaries are hundreds of meters deep (and more) and are not located in the regions analysed. It should be mentioned what are the tracers that will be downscaled at the boundaries? Salinity is certainly one of them, why was is not included in the analysis?

Main text

Line 71: why look only at three variables? What about salinity?

Line 78: historical simulations are not used for projections. This should be rephrased.

Lines 115-116: The ESMs horizontal resolution in the region of interest should be given in Table 1. Some models have a variable resolution and it might not be that bad in the NWA .

Line 117: MR and HR mean medium resolution and high resolution respectively. If they share the same grid where does the change in resolution come from?

Line 123: From where were the satellite data obtained. Who did the averaging?

Line 128: Which data from the AZMP were used? Along the Halifax line only? Why were the data averaged seasonally and not monthly like the other data?

Line 132: So the model results are brought back onto 3 different grids, one for each

variable. Are the time period also adjusted? For example SST goes from 2006 to 2016. The CMIP5 historical period ends in 2005. How can the two be compared then? Also, there are probably models that have a higher resolution than 1° (see my comment for lines 115-116), what is the impact of decreasing the resolution (converting from higher to lower resolution) and having on the ranking analysis. And how is the conversion of one grid to the other done? Also, how the thickness of the first grid cell compares between the different models?

Line 175: What is the main difference between the CMIP5 and CMIP6 groups, why is it better? Improved BGC? Same question hold for nitrate.

Line 200: Figure 6 does not show the annual cycle.

Line 208: Could you explain why? From local atmospheric forcing or circulation change?

Line 209: What are the improvements in the CMIP6 models?

Line 215: So this means that the model ranked 2 (and others)might not be ranked as high? What is the impact on the final choice?

Line 218: Could the fact that you are using different time periods and different grid resolution for the three variables explain the lack of correlation?

Line 221: How do you explain that?

Line 230: Why? Does it relate to temperature-dependant phytoplankton growth?

Line 245: What are the years compared for the ACM and the glider data?

Line 260: Correlation coefficients are high for nitrate despite having a large bias and RMSD. This should be explained.

Line 270: See my previous comments about time-period and grid differences. I think that a statement about a misrepresentation of ocean physics as the cause should be

backed up since later on the cause for nitrate mismatch is stated as coming from the BGC behavior (line 279). There are refs for the CMIP5 models but was there an improvement in circulation with the CMIP6 group or not?

Line 288: So here again, the model-data comparison was made on a different grid than for the ESMs. Shouldn't it be done on the same grid for an appropriate comparison?

Line 295: Lavoie et al. (2019) also point at the misrepresentation of the remineralisation depth in those models as a likely cause. This also explain why some models having a coarse resolution still have good results with biogeochemistry. But the statement made below that improving the model resolution does not improve the representation of circulation and main features in the models, such as the representation of the Gulf Stream detachment point and flow around the Grand Banks should be demonstrated. There is a large consensus on that and it should not be stated lightly. The authors could actually show the mean currents between the two versions of a same model with improved resolution. Especially that you state that higher resolution is required to refine the projections on line 306. There is a contradiction here.

Line 310: Here again it appears to be contradictory as you previously mentioned that BGC improvements we the cause for improvements in the CMIP6 ranking. There are likely different versions of the 4 BGC models mentioned, which should be specified in the table and considered in the analysis. Also, it could relate to the processes that control nitrate in the regions under study, they are different for your 3 regions. And how well are these processes represented by the ESMs?

Line 335: What was updated in the ocean biogeochemistry?

Figure 4f: In the suppl. figures, there is more chl a in the model than in the obs. The opposite is shown here.

Figure 7 : Maybe use ACM instead of ROMS.

Figure 8: Could specify that ACM has the same rank for the three variables (only see

one point)

---

## Author Comment (AC2) · 2 Oct 2020

**Detailed responses to reviewer 2** (reviewer comments are included in black, responses in blue font)

**General comments**

**Comment:**
1. In this paper, the authors compare the output of CMIP5 and CMIP6 Earth System Models (ESMs) to observations in order to determine which models are suitable to build boundary conditions for projections. A ranking analysis was performed on a large array of ESMs. However, they are only looking at surface values of 3 variables and far away from the regional model boundaries, even though they mention on lines 44-46 that it is important to look at the information imposed at the boundaries. I think the objective stated on line 67 "Our objective is to assess the performance of a number of available ESMs in reproducing present conditions on the NWA shelf in contrast to a high-resolution regional model" is more in line with what is presented in the manuscript since there is no analysis at the boundaries.

*Response:* We agree with the reviewer that ESMs performance offshore, where the regional model boundary is located, is most important for regional downscaling and may differ from those on shelf, although we suspect that performances in/out of the shelf are related. To address this specific point that was also raised by reviewer 1, we will add an analysis of ESM performances along the ACM boundaries and compare those with the results from the shelf.

As mentioned in the response to reviewer 1's general comments, we will also clarify the objectives and findings in the revised manuscript and provide further discussion about the regional use of ESM data. For example, ESM projections can be used to drive higher trophic level models and to assess societal impacts of climate change, such as fish catch, that affect mainly Exclusive Economic Zones (EEZ), i.e. coastal ecosystems. Our results indicate that the choice of ESM for these projections is very important.

**Comment:**
2. They are not discussing the processes that lead to the observed values in the region under study and they are not analysing if the models do represent these processes correctly. I believe salinity should be included, as surface temperature depends strongly on atmospheric forcing while salinity is more representative of the different water masses in that region.

*Response:* The study is meant to provide ESM users with information about model performance for either direct use or regional downscaling. We do discuss to some extent the potential sources of mismatch between models and observations but 1) this is not the objective of the analysis and 2) we can only speculate on the sources of errors.

We included temperature in the comparison because it is an important variable for higher trophic level studies and climate change impacts. The fact that surface temperature is available at high spatial and temporal resolution on the shelf, similar to chlorophyll, is also important. Despite the tight control by atmospheric forcing, we did find significant

differences in surface temperature across the ESMs. We believe these differences are of interest and relevant to many users.

Large scale salinity patterns in the historical simulations are likely related to those of temperature and therefore it is not clear if adding salinity to the comparison would provide additional information. However, since salinity is available from the WOA dataset at the same resolution as $NO_3$, we will compare simulated and observed salinity and add the results to the manuscript.

**Comment:**
3. Moreover, it is very surprising that a similar study (Lavoie et al. 2019) in the exact same area, with the same purpose, and using some of the same ESMs is hardly mentioned at all. No comparison of the results of this study with the 2019 study is made.

*Response:* We agree with the reviewer that we should discuss our results with respect to the findings of Lavoie et al. (2019). We did mention the conclusions of an earlier report (Lavoie et al., 2015) but will expand this discussion and also add the more recent study to the revised manuscript.

**Comment:**
4. Also there is not enough details on the comparison with the data, they appear to be comparing different time periods (see detailed comments) or on how the ESMs were brought to a single grid.

*Response:* We provide the information about time range, averaging and spatial mapping in the Methods. Additional information will be added for completeness, as detailed in the responses to detailed comments 18–20, 27 and 30 below.

**Comment:**
5. There is only a vague mention of what the improvements are between the CMIP5 and CMIP6 models. What was improved should be stated (not only biogeochemistry of physics) so that the reader can judge on the potential impact on the ranking.

*Response:* See response to comment 2 above and comment 25 below. Model changes from CMIP5 to CMIP6 can be significant, including in the atmospheric and terrestrial realms, and the study is not meant to find out what are the sources of improvement in performances. Give the limited output available from these models, we can only speculate on the sources of improvement based on our results. The reader is referred to the specific papers listed in Table 1 for the list of changes in the models. To clarify this point we will add the following statement L317:

"*For specific changes in the CMIP6 model versions, the reader is referred to the references listed in Table 1.*"

**Comment:**
6. Increasing the model resolution in order to improve the representation of the circulation in the NWA has been mentioned by many authors (e.g. Loder, Brickman, Yool). Here it is stated that the resolution does not have an impact. This is a big

statement, considering the general agreement, and it should be demonstrated. The authors could show the changes in circulation of a few models they are giving in example for this.

*Response:* Our findings are in line with previous work, included the ones cited above, see responses to detailed comments 9, 16, and 34–35.

**Comment:**
7. All these points should be addressed in order for the conclusions to be more convincing (ranking based on analysis of shelf surface conditions representative of boundary conditions).

*Response:* These points are addressed in the detailed comments below.

**Comment:**
8. Also, Lavoie et al. (2019) estimated that the boundary conditions obtained with the ESMs were not as reliable for the simulation of the conditions on the Scotian Shelf and in the Gulf of Maine. It would good to know if there was an improvement in this regard with the CMIP6 ESMs.

*Response:* In the revised manuscript we will provide an analysis of model performance along the ACM boundaries. The change in ranking from CMIP5 to CMIP6 along the western, southern and northern boundary will show if there was an improvement in the CMIP6 models. We will discuss these new results with respect to the findings of Lavoie et al. (2019).

**Specific comments**

**Comment:**
9. Line 11: Here you say that the coarse resolution is not appropriate to represent the circulation and elemental flux but later on you say that increasing the resolution does not matter. Is it important or not?

*Response:* Our two statements are in agreement. It is well known that the coarse resolution of ESMs is an issue to resolve shelf-scale processes and that high resolution is necessary in these areas, as mentioned in comment 6 above. However, even the highest resolution ESM from our ensemble is too coarse to resolve shelf-scale processes and therefore it is not surprising that we do not see better performance with increasing ESM resolution. We will clarify this point by adding the following sentence Line 305:

"*The lack of correlation between model resolution and performance on the NWA shelf is not surprising as all ESMs are coarse and do not explicitly resolve shelf-scale processes but rather rely on their parameterisation. Much higher resolution will be necessary...*"

**Comment:**
10. Line 14: ability to reproduce surface observations...

*Response:* Will be corrected.

**Comment:**
11. Line 15: why is it particularly sensitive?

*Response:* We refer to the effect of climate change on the location and strength of the Gulf Stream and Labrador Sea currents. We will clarify the sentence in the revised manuscript.

**Comment:**
12. Line 16: The spatial mismatch in large-scale circulation was not demonstrated. There are references for CMIP5 but what about CMIP6. Changes, or not, in circulation should be shown/mentioned after an inspection of the ESMs results.

*Response:* We mentioned a warm bias in the Gulf of Maine that is in line with the results of Loder et al. (2015) and Saba et al. (2016) (Lines 365-266). Although smaller, a cold bias appears on Grand Banks in most models (Figures 4c and 5c). The biases suggest a mismatch in the large-scale currents. However, since we did not compare the position of the currents across the models, we will rephrase the sentence L15–17 as follows:

"*Most ESMs compare relatively poorly to observed nitrate and chlorophyll and show differences with observed temperature that suggest a spatial mismatch in their large-scale circulation.*"

Since we will add salinity and look at offshore conditions along the ACM boundaries we will have more support for this statement.

**Comment:**
13. Line22: How can we say just by looking at the surface temperature, nitrate and chl a that the top three models are appropriate for boundary forcing? The model boundaries are hundreds of meters deep (and more) and are not located in the regions analysed. It should be mentioned what are the tracers that will be downscaled at the boundaries? Salinity is certainly one of them, why was is not included in the analysis?

*Response:* The revised manuscript will include both salinity and a comparison along the offshore boundaries of the ACM, see responses to comments 2 and 8 above, and response to comment 1 by reviewer 1.

**Comment:**
14. Main text Line 71: why look only at three variables? What about salinity?

*Response:* Salinity will be included, see response to comment 2 above. Not all variables were available for all models (ESMs and ACM) and could be compared to observations so we restricted the comparison to 3 variables, plus salinity in the revised manuscript. The selected variables are, arguably, the most important to potential users.

**Comment:**
15. Line 78: historical simulations are not used for projections. This should be rephrased.

*Response:* TBA.

**Comment:**

16. Lines 115-116: The ESMs horizontal resolution in the region of interest should be given in Table 1. Some models have a variable resolution and it might not be that bad in the NWA.

*Response:* To give a sense of horizontal resolution that is easily comparable across models we provide the number of grid cells in the three zones of interest in Table 1. This value also depends on the coverage, which can be very poor for coarse grids (e.g. IPSL–CM5), and therefore provides more information to the reader. However, as suggested by the reviewer and since resolution is typically reported in degrees for ESMs, we will add a column to Table 1 with average $\Delta$lon $\times$ $\Delta$lat on the NWA shelf.

**Comment:**

17. Line 117: MR and HR mean medium resolution and high resolution respectively. If they share the same grid where does the change in resolution come from?

*Response:* MR stands for Mixed Resolution and HR for Higher Resolution in the MPI model names. MPI-ESM-MR (CMIP5) and MPI-ESM1-2-HR (CMIP6) have the same ocean circulation model but the horizontal resolution of the atmospheric component was improved from ~200 km (MPI-ESM-MR) to ~100 km (MPI-ESM1-2-HR). Thus, model names are not related to the ocean model, which can be confusing. A similar confusion can occur from the IPSL CMIP5 model names. In this case, the models share the same ocean model, but the horizontal resolution of the atmospheric model is higher in the medium resolution (MR) version compared to the low resolution (LR) version. To avoid some confusion, the following text will be added to the caption of Table 1:

"*Note that the IPSL-CM5 models share the same ocean component with higher resolution atmospheric component in the MR version. Similarly, MPI-ESM-MR and MPI-ESM1-2-HR share the same ocean component with higher resolution atmospheric component in the HR version.*"

**Comment:**

18. Line 123: From where were the satellite data obtained. Who did the averaging?

*Response:* Links to the data will be added as follows:

"*1) satellite surface chlorophyll observations from the Sea-viewing Wide Field-of-view Sensor (SeaWiFS) as 8-day averaged maps at 1/12˚ resolution (1999–2010, https://doi.org/data/10.5067/ORBVIEW-2/SEAWIFS/L3M/CHL/2018), 2) surface nitrate from the World Ocean Atlas 2009 (WOA; Garcia et al., 2010) at 1˚ resolution, and 3) surface temperature from the Operational SST and Sea Ice Analysis (OSTIA) system (Donlon et al., 2012) at 1/20˚ resolution (2006–2016, https://doi.org/10.5067/GHOST-4FK01). Monthly climatologies were calculated for each of these.*"

The following references will be added:

SeaWiFS. NASA Goddard Space Flight Center, Ocean Ecology Laboratory, Ocean Biology Processing Group. Sea-viewing Wide Field-of-view Sensor (SeaWiFS) Chlorophyll Data; NASA OB.DAAC, Greenbelt, MD, USA. doi:10.5067/ORBVIEW-2/SEAWIFS/L3M/CHL/2018. Accessed on 2014/03/12.

OSTIA. UK Met Office. 2005. GHRSST Level 4 OSTIA Global Foundation Sea Surface Temperature Analysis. Ver. 1.0. PO.DAAC, CA, USA. doi:10.5067/GHOST-4FK01. Accessed on 2019/12/06.

The original data were daily (OSTIA) and 8-day (SeaWiFS) maps which were converted to monthly climatologies, as mentioned Lines 126–127.

**Comment:**
19. Line 128: Which data from the AZMP were used? Along the Halifax line only? Why were the data averaged seasonally and not monthly like the other data?

*Response:* See also response to comment 30. Yes, along the Halifax Line where both high-resolution glider data and ship-based bi-monthly or seasonal data are available. The location of the data is presented in Figure 1. We will rephrase the sentence as follows for clarity:

"*In addition, the regional model was validated using high-resolution in-situ observations along the Halifax Line (Figure 1) from the Atlantic Zone Monitoring Program (AZMP, 2000–2014, http://www.meds-sdmm.dfo-mpo.gc.ca/isdm-gdsi/azmp-pmza/index-eng.html) and glider transects between 2011 and 2016 (Ross et al., 2017)*"

The glider missions and the AZMP data collection frequency along the Halifax Line were seasonal, which is why the spatially resolved dataset was averaged into seasons rather than months. At station 2 we were able to use a bi-weekly frequency for the AZMP climatology.

**Comment:**
20. Line 132: So the model results are brought back onto 3 different grids, one for each variable. Are the time period also adjusted? For example SST goes from 2006 to 2016. The CMIP5 historical period ends in 2005. How can the two be compared then? Also, there are probably models that have a higher resolution than 1◦ (see my comment for lines 115-116), what is the impact of decreasing the resolution (converting from higher to lower resolution) and having on the ranking analysis. And how is the conversion of one grid to the other done?

*Response:* We used a heterogeneous data set and for comparison we brought the data and model to the same temporal (monthly) and spatial (observation grid) scale. We used a long-term climatology for robustness. Ideally, we would use the same time range for observations and models but this was not possible. All the ESMs used the same time range (30 years climatology, 1976–2005) so their intercomparison is robust. Note that Line 193 should read "(1976–2005)", not "(1975–2005)", which will be corrected in the revised manuscript. Unfortunately, we could only run the ACM simulation for 15 years starting in 1999 so the ESMs and ACM simulations overlap for 6 years only. Since the

CMIP6 historical simulations end in 2014 it was possible to use the range 2000–2014 with the CMIP6 models. However, the ESM intercomparison would have been less robust and we decided to use the same time range for all the ESMs.

The conversion from grid to grid is simply a linear interpolation onto the observations grid. This information will be added Line 133 as follows:

"*For comparison with the observations, each model was mapped onto the SeaWiFS, WOA and OSTIA grids using a linear interpolation*"

**Comment:**
21. Also, how the thickness of the first grid cell compares between the different models?

*Response:* The ESMs have various vertical resolution. For completeness, the number of vertical levels will be added to Table 1. The thickness of the vertical layers may influence the model performance but this is inherent to the model configuration and therefore not relevant here.

**Comment:**
22. Line 175: What is the main difference between the CMIP5 and CMIP6 groups, why is it better? Improved BGC? Same question hold for nitrate.

*Response:* In the discussion (see L329–341) we speculate about the source of improvement in surface chlorophyll and nitrate fields. The suggested sources of improvement refer to the literature as we cannot substantiate the reasons for these changes from our data.

**Comment:**
23. Line 200: Figure 6 does not show the annual cycle.

*Response:* Here we refer to the data that are used to calculate the RMSD. The sentence will be modified to:

"some models are much better at representing the observed annual cycle, as indicated by the lower RMSD (Figure 6)".

**Comment:**
24. Line 208: Could you explain why? From local atmospheric forcing or circulation change?

*Response:* We can only speculate the reason why some models have poor scores for temperature. Lines 264–266 we mention the warm bias associated with a mismatch in the location of the Gulf Stream. However, we do not know why some CMIP6 models (CESM and GISS) have a large temperature bias. These models already had poor scores for temperature in their CMIP5 version. Line 323 the following sentence will be added:

"*Models with poor scores had already poor scores in their CMIP5 version and therefore the cause of their poor performance is likely the same.*"

**Comment:**
25. Line 209: What are the improvements in the CMIP6 models?

*Response:* Here "improvement" refers to the lower chlorophyll scores for the CMIP6 models 22 and 23 (CNRM-ESM2-1 and GFDL-ESM4). These models have the best chlorophyll scores after ACM. For clarity, "improvement" will be removed and the sentence will be:

"*The range of variability in chlorophyll scores did not reduce from CMIP5 to CMIP6 and given the relatively low scores of a few CMIP6 models (i.e. 22 and 23), the range is larger in the CMIP6 group (0.8–1.4, Figure 7, right panel) than in the CMIP5 group (1–1.4, Figure 7, left panel).*"

**Comment:**
26. Line 215: So this means that the model ranked 2 (and others) might not be ranked as high? What is the impact on the final choice?

*Response:* The numbers in the parenthesis correspond to the model ID. Since we excluded Nov–Jan from the WOA dataset, i.e. when nitrate is high at the surface, models with consistently low nitrate will have lower scores than they should, which will increase their rank with respect to nitrate. The overall rank is an average of the 3 variables so it should be less sensitive to this effect. Supporting Figures S1-S5g-i indicate that this might be the case for models 4, 8, 14, 19, and 26–27. Models 4, 8 and 19 have poor rankings so the underestimation of the nitrate score have no effect on the final ranking. However, for model 14, 26–27, information on the underestimation of the nitrate scores should be provided. To reflect a possible bias in the ranking of these models we will add an asterisk in Table 2 beside the overall rank of these models and will update the caption accordingly.

**Comment:**
27. Line 218: Could the fact that you are using different time periods and different grid resolution for the three variables explain the lack of correlation?

*Response:* We used climatologies to remove as much as possible the influence of time on the comparisons. For the sake of completeness, in the supplement we will provide 1:1 comparison of observed chlorophyll, nitrate and temperature on each grid and refer to the comparisons in the revised manuscript.

**Comment:**
28. Line 221: How do you explain that?

*Response:* We do not wish to speculate about the reasons for each models' individual ranking. As stated before, given the limited output that is available for each of the models we would have to speculate but this is outside the intended scope of this study. The objective is to report on the models' behaviour.

**Comment:**
29. Line 230: Why? Does it relate to temperature-dependant phytoplankton growth?

*Response:* Again, we can only speculate. Temperature-dependant phytoplankton growth is a possibility, large scale circulation is another. If poor chlorophyll scores also correspond to poor salinity scores in our new results then large scale circulation might be a better explanation. We will include this discussion in section 4.1 of the revised manuscript.

**Comment:**
30. Line 245: What are the years compared for the ACM and the glider data?

*Response:* Information on the ACM and glider data is provided in the Methods, i.e. Lines 111 and 130. The ACM data are the same as for the comparison with the ESMs, i.e. years 2000–2014 but presented as a seasonal (Figure 9) and daily (Figure 10) climatology to match the resolution of the glider data. The AZMP years are the same as ACM. The glider missions were carried out between 2011 and 2016 but were heterogeneous in time and space (see tracks on Figure 1). To enable a quantitative comparison between the glider and ACM data (Table 3), we spatially interpolated both dataset onto a transect following the Halifax Line (black line in Figure 1). The glider missions were seasonal, which is why the spatially resolved dataset was averaged into seasons (Figure 9). For each mission, data were extracted at Station 2 to produce a monthly climatology (Figure 10). ACM data were extracted at this location for comparison. We will add this information as follows in the Methods section.

**Comment:**
31. Line 260: Correlation coefficients are high for nitrate despite having a large bias and RMSD. This should be explained.

*Response:* The correlation coefficient is a complementary measure to bias and RMSD. Correlation and bias are largely unrelated. The former is a measure the similarity in spatial or temporal variations but does not account for bias. In other words, the same correlation coefficient can occur for very different values of bias. Likewise, high correlation does not imply low RMSD. In a noisy data set the RMSD will be higher than in a data set that is smooth, while both might display the same correlation.

**Comment:**
32. Line 270: See my previous comments about time-period and grid differences. I think that a statement about a misrepresentation of ocean physics as the cause should be backed up since later on the cause for nitrate mismatch is stated as coming from the BGC behavior (line 279). There are refs for the CMIP5 models but was there an improvement in circulation with the CMIP6 group or not?

*Response:* The mismatch is partly associated with ocean physics and partly due to the BGC model. To reflect this the sentence will be modified as follows:

"*The correlation between temperature and chlorophyll scores indicated that errors in surface chlorophyll concentration were partly driven by the misrepresentation of the general circulation and, more generally, of ocean physics. The improvement in chlorophyll from CMIP5 to CMIP6 without an associated improvement in temperature*

*suggest that the errors in surface chlorophyll were also driven to some extent by a poor biogeochemical model component*".

**Comment:**
33. Line 288: So here again, the model-data comparison was made on a different grid than for the ESMs. Shouldn't it be done on the same grid for an appropriate comparison?

*Response:* No, the grid for comparison depends on the dataset. Here, since we have both high (glider) and low (AZMP) spatial resolution data, we mapped the data along the Halifax Line.

**Comment:**
34. Line 295: Lavoie et al. (2019) also point at the misrepresentation of the remineralisation depth in those models as a likely cause. This also explain why some models having a coarse resolution still have good results with biogeochemistry. But the statement made below that improving the model resolution does not improve the representation of circulation and main features in the models, such as the representation of the Gulf Stream detachment point and flow around the Grand Banks should be demonstrated. There is a large consensus on that and it should not be stated lightly. The authors could actually show the mean currents between the two versions of a same model with improved resolution. Especially that you state that higher resolution is required to refine the projections on line 306. There is a contradiction here.

*Response:* See also response to comment 9 above. We agree with the reviewer that there is a large consensus on the effect of grid resolution on large scale circulation and our discussion is in line with this consensus. The resolution of the CMIP models is much coarser than the resolution of the models used to study the effect of grid resolution on the large-scale current systems of the NWA. Therefore, as pointed out by reviewer 1 (see comment 7 by reviewer 1), it is not surprising that current ESMs do not show the effect of grid resolution on model performances; much higher resolution will be necessary to see this effect. We will clarify this point as follows after Line 305:

"*The lack of correlation between model resolution and performance on the NWA shelf is not surprising as all ESMs are coarse and do not explicitly resolve shelf-scale processes but rather rely on their parameterisation. Much higher resolution will be necessary…*"

We will also add the following sentence Line 295:

"*In the NWA, Lavoie et al. (2019) suggest that the misrepresentation of remineralisation depth may lead to poor results in some models, despite their resolution.*"

**Comment:**
35. Line 310: Here again it appears to be contradictory as you previously mentioned that BGC improvements we the cause for improvements in the CMIP6 ranking. There are likely different versions of the 4 BGC models mentioned, which should be specified in the table and considered in the analysis. Also, it could relate to the processes that control nitrate in the regions under study, they are different for your 3 regions. And how well are these processes represented by the ESMs?

*Response:* Here we refer to the general BGC component. There are not enough data to compare specific model version or parameterization. This paragraph is meant to point out that, in our comparison, there was no relationship between the type of model and the overall performances. But we cannot go further, and this is not the objective of the study. For clarification we will modify the paragraph as follows:

"*Although model performance is likely influenced by the biogeochemical model structure, we did not find a clear relationship between the type of biogeochemical model and performance. Here we only refer to the model type because the same model may have different parameterizations when used by different groups. While the inner and outer ensembles share only 4 biogeochemical models (PISCES, HAMOCC, TOPAZ2, NOBM) out of 13, there was no indication of consistently better performance for the biogeochemical models in the inner ensemble. For example, models using similar ocean biogeochemistry (e.g., PISCES: 5, 12–14 (CMIP5), 22 and 26 (CMIP6), and HAMOCC: 15–16, 18 (CMIP5), 28–29 (CMIP6)) had very different ranks, with no obvious relationship between overall model rank and the ocean biogeochemical model component.*"

**Comment:**
36. Line 335: What was updated in the ocean biogeochemistry?

*Response:* As mentioned in the response to previous comments above, our goal is not to discuss the details of the models. The HAMOCC biogeochemistry module includes a new parameterization of detritus sinking, which may influence surface chlorophyll and nitrate, as suggested by Lavoie et al (2019). However, this explanation is highly speculative and we do not think that is should be included here.

**Comment:**
37. Figure 4f: In the suppl. figures, there is more chl a in the model than in the obs. The opposite is shown here.

*Response:* The supplemental figures S1–S5 present the individual chlorophyll time series for the 29 ESMs, whereas Figure 4f shows ECM ensembles. The "all" ensemble time series are calculated with all the individual ESM time series in Figures S1–S5. Figure 4f shows that even though individual ESMs can be close to observations during the spring bloom (e.g. HadGEM2, Figure S2f) and even significantly larger (e.g. CESM2, Figure S4f), all ESM ensembles underestimate the bloom.

**Comment:**
38. Figure 7: Maybe use ACM instead of ROMS.

*Response:* Will do.

**Comment:**
39. Figure 8: Could specify that ACM has the same rank for the three variables (only see one point)

*Response:* We will add the following sentence to the caption:

*"Hidden coinciding ranks (models 1, 6, 30 and 18) are provided in Table 2."*

---

## Author Response (AR1)

**Detailed responses to reviewer 1** (reviewer comments are included in black, responses in blue font)

**Overview**

The study addresses the important question of whether the Earth system models can accurately simulate biogeochemical processes/fields on the Northwest North Atlantic shelf and if they are suitable for projecting the regional impact of climate change. The study ranks the CMIP5 and CMIP6 models in terms of capturing the observed seasonal variability in surface temperature, chlorophyll and nitrate in the Northwest North Atlantic shelf, highlighting that ESMs are not adequate to project accurately future changes in the NWA shelf. Based on the comparison with these ocean surface observations the study clearly demonstrates how a regional model outperforms the Earth system models in terms of capturing the seasonal variability in the region during the historical period. The study then provides plausible reasons for the better performance of some of the Earth system models and the improvement from the CMIP5 to CMIP6.The study is interesting, timely and I believe it is a substantial contribution towards improving our understanding and ability to provide accurate regional projections of climate change on the Northwest North Atlantic shelf, particularly in the context of biogeochemical fields. The methods and results are reasonable, justified and well presented. Hence, in my opinion the study fits well the Biogeosciences scope and should be consider for publication after addressing some very minor comments/revisions as I explain below.

*Response:* We appreciate the positive and constructive feedback. We provide detailed responses to the comments below.

**General comments**

**Comment:**
1. I agree that not all the ESMs are suitable to force regional model projections and should be selected carefully as not to force the regional model with unrealistic large-scale patterns. The choice of the 'parent model' should be based on its performance on capturing the large scale patterns/features that are introduced in the regional models such as: large scale circulation patterns, and biogeochemical and physical fluxes  in-out of the regional model domain (the study highlights well this point in the introduction, lines 48-59 and in discussion, lines 343-351). Misrepresentation of the ESMs surface fields in the Gulf of Maine, Scotian Shelf and Grand Banks may be due to the models not being able to accurately represent on-shelf fine-scale biogeochemical and physical processes due to lack of resolution (which is inadequate to resolve the physical/biogeochemical processes on the shelves in all the ESMs, as mentioned in the manuscript), lack of accurate river discharge, biogeochemical model parameter optimisation for global use rather than the region of interest, or lack/misrepresentation of other local processes (e.g., enhance mixing associated with tides). The regional model is meant to address the above on-shelf problems, and as long as the large-scale patterns are introduced correctly at the open boundaries it should be able to perform well. Hence, I am a little reserved about the conclusion that 3 specific ESMs are most appropriate to force regional model projections in NWA (lines 21-23 and 77-79) based on the ESMs' performance on the shelf at the surface. In my understanding, there is no explicit analysis in the study to link the ESMs

ranking on the shelf with the large-scale physics/circulation patterns off-shelf, or across-shelf properties exchange beyond a moderate correlation of surface temperature and surface chlorophyll, and no explicit link between the ESMs performance on the shelf and their performance off the slope in the vicinity of the regional model boundaries. So in my opinion, I would suggest to further discuss and clarify this issue: does low ranking on the shelf translate directly to low ranking of the larger scale patterns in the region and specifically to low ranking in the quality of the ESMs exchange/input of physical and biogeochemical properties at the location of the open boundaries offshore the Scotian shelf, Gulf of Maine and Grand Banks? Why are the ESMs with high ranking on the shelf the ones that can provide the best open boundary conditions coming off the shelf?

*Response:* We agree with the reviewer that ESMs performance offshore, along a regional model boundary such as the ACM, is most important for regional downscaling and may differ from those on shelf, although we suspect that the performance on the shelf is related to the performance along the boundaries. To address this specific point we now provide an additional analysis of ESM performances along the ACM boundaries and compare those with the results from the shelf, see supporting Figures S10–13 and the following text on lines 272 to 281:

"*Model scores and ranking were also calculated along the boundaries of the regional model (see supporting Figure S10). The ranking shows that model performance on the shelf is not necessarily indicative of the performance along the boundaries of the regional model (supporting Figure S11, Table S2). Moreover, individual rankings are much more variable at the boundaries, even for the best performing models. The 8 best ESMs along the boundaries (22, 11, 30, 28, 16, 10, 26, 6) have an average rank of 9.2– 10.5. There are no significant correlations between individual rankings, including temperature and salinity. Nonetheless, there is some agreement between the shelf and the outer boundary ranking for chlorophyll ($\rho="0.80"$ ), nitrate ($\rho="0.81"$ ) and salinity ($\rho="0.81"$ , supporting Figure S12 and Table S3). Interestingly, the agreement is better with CMIP6 models (Table S3). However, there is no agreement for temperature. A similar pattern is found for individual boundaries (Figure S13). In this case, and apart from temperature, the model ranks along the northeastern boundary agree the most with those from the shelf.*"

We also clarified the objectives and findings of our analysis and provide further context about regional applications using ESM output, which is not only used to provide boundary conditions for regional downscaling. For example, ESM projections are used to drive higher trophic level models and to assess societal impacts of climate change, such as fish catch, that affect mainly Exclusive Economic Zones (EEZ), i.e. coastal ecosystems. Our results indicate that the choice of ESM for these projections is very important. We added the following text on lines 60 to 64:

"*Despite these issues, CMIP historical simulations and future projections have been used to characterize biological responses to climate change in the NWA (e.g., Bryndum-Buchholz et al., 2020a; Greenan et al., 2019; Lavoie et al., 2019; Stortini et al., 2015; Wilson et al., 2019; Wilson and Lotze, 2019). ESM selection in these regional studies is either qualitative or based on either scenario outcomes (e.g. variability across models) or*

*global assessments rather than on regional model performance. However, ESMs that poorly represent the dynamics of the NWA will affect the results of regional studies.*"

**Specific comments**

**Comment:**
2. Line 75 (typo): I think an 'on' is missing such that: 'based on the mismatch...'

*Response:* Corrected.

**Comment:**
3. Lines 147-148 (just a suggestion): The authors can refer and link R and Rbio symbols in Table 2 directly with the rank for temperature, chlorophyll and nitrate, and the rank of chlorophyll and nitrate only, respectively, to help the readers follow more easily the ranking metric.

*Response:* Thank you for the suggestion, we now refer to $\bar{R}$ and $\bar{R}^{bio}$ as follows:

L160-162: "*The overall rank was determined by ranking models by the averages of their ranks for surface temperature, salinity, chlorophyll, and nitrate ($\bar{R}$). For models with equal averages the ranking was determined by the average of chlorophyll and nitrate ranks ($\bar{R}_{bio}$).*"

**Comment:**
4. Line 179: Please can you clarify which months you consider for the winter bias for nitrate? In my understanding winter is defined as December-February (based on Table3) but November-January are excluded from the observations for nitrate, so is it only February?

*Response:* Nov–Jan were excluded from the World Ocean Atlas (open/close circles in Fig 5g-i) and therefore in the Gulf of Maine there are only February data for the winter. However, for the Scotian Shelf and Grand Banks AZMP data were also available (black squares in Fig 5h-i) and are included in the winter (December-February) bias calculation. This was be clarified as follows in the captions of Figures 4 and 5:

"*Nov–Jan WOA nitrate data are excluded (open circles). Model comparison with observations in the Gulf of Maine is therefore only available from February to October. For the Scotian Shelf and the Grand Banks additional AZMP data are available. In case of multiple observations, the data are monthly averaged.*"

And in the main text as follows:

L199-200: "*Note that since Nov.–Jan. nitrate WOA observations were excluded from the analysis (see section 2.3), winter observations are only available in February in the Gulf of Maine and in December and January in Grand Banks.*"

**Comment:**
5. Line 233 (just a suggestion): Maybe if you could clarify that here you mean the best overall ESM in terms of nitrate and chlorophyll (the Rbio in Table 2) by adding

something along the lines: 'ACM and model 22 (the best overall ESM for combined nitrate and chlorophyll) indicates...'.

*Response:* The sentence was changed to:

L263-264: "*The gap between ACM and model 28 (ESM with best $\bar{R}$ and $\bar{R}_{bio}$, Table 2) indicates that none of the ESM performs best for all fields, especially for both chlorophyll and nitrate.*"

Note that the nitrate field was reprocessed for model 22 after we found an error in the original file and its nitrate score degraded slightly. The best ESM is now model 28.

**Comment:**
6. Lines 268-270: I agree that some of the errors in chlorophyll are linked with the temperature bias and the misrepresentation in general circulation. However, in my opinion i) the relative moderate correlation between chlorophyll and temperature r=0.51 and ii) the improvements in chlorophyll in CMIP6 relative to CMIP5 models that are not associated with any improvement in temperature (as discussed in lines 321-328) indicate that the errors in surface chlorophyll are also driven to a significant degree by a poor biogeochemical model component rather than by the ocean physics only. Hence, I suggest that the authors could modify this sentence to reflect this.

*Response:* This paragraph was modified as follows:

L311-322: "*The correlation between temperature and chlorophyll scores (and to a lesser extent salinity) and the concomitant poor scores in chlorophyll and temperature/salinity (i.e. Group C in Figure 7) indicate that errors in surface chlorophyll concentration are partly driven by a misrepresentation of the general circulation and, more generally, of ocean physics. The improvement in chlorophyll from CMIP5 to CMIP6 without an associated improvement in temperature suggest that the errors in surface chlorophyll were also driven to some extent by errors in the biogeochemical model component. Lavoie et al. (2019) indicated that the misrepresentation of primary production in the NWA may be associated with the misrepresentation of particulate organic matter sinking and remineralization in the subsurface layer. They found an annual subsurface nitrate peak in CanESM2, GFDL-ESM2M, NorESM1-ME, CESM1-BGC (models 2, 7, 18 and 3, respectively) similar to the high surface nitrate found in this study (supporting Figures S1 and S3). However, all these models had poor scores in our assessment and therefore do not provide an appropriate representation of the biogeochemistry on the NWA shelf (Figure 8) or along the ACM boundaries (Figure S11). However, it is not possible, and beyond the scope of this work, for us to draw conclusions about the source of the regional mismatch in surface chlorophyll and nitrate in the ESMs.*"

**Comment:**
7. Section 4.2: Although the authors mention it, in my opinion, they could highlight even more that the lack of correlation between resolution and rank is not surprising as none of the ESMs have the resolution to explicitly resolve the processes in shelf-scales rather than parameterise them. Maybe the authors could add a sentence after line 303 along the lines: 'This lack of correlation between model resolution and accuracy on the NAW shelf

and the primary control of performance by the model set-up is not surprising as all ESMs are coarse and do not explicitly resolve the shelf-scale processes but rather rely on their parameterisation.

*Response:* The following sentence was added:

L390-392: "*The lack of correlation between model resolution and performance on the NWA shelf is not surprising as all ESMs are still coarse and do not explicitly resolve shelf-scale processes but rather rely on their parameterisation. Much higher resolution will be necessary…*"

**Comment:**
8. Figure 8 caption (just a suggestion): The authors mention that the temperature rank is hidden for model 6, however, in my understanding ranks for other models are also hidden as they coincide (model 1, 30 and 18). Maybe just for clarity and to make the link between figure 8 and table 2 explicit, you could mention in the caption something along the lines 'Coinciding ranks as shown in table 2 are hidden'.

*Response:* The following sentence was added to Figure 8 caption:

L805: "*Hidden coinciding ranks (models 2, 3, 6, 10, 11, 18, 27, 28 and 30) are provided in Table 2.*"

**Detailed responses to reviewer 2** (reviewer comments are included in black, responses in blue font)

**General comments**

**Comment:**
1. In this paper, the authors compare the output of CMIP5 and CMIP6 Earth System Models (ESMs) to observations in order to determine which models are suitable to build boundary conditions for projections. A ranking analysis was performed on a large array of ESMs. However, they are only looking at surface values of 3 variables and far away from the regional model boundaries, even though they mention on lines 44-46 that it is important to look at the information imposed at the boundaries. I think the objective stated on line 67 "Our objective is to assess the performance of a number of available ESMs in reproducing present conditions on the NWA shelf in contrast to a high-resolution regional model" is more in line with what is presented in the manuscript since there is no analysis at the boundaries.

*Response:* We agree with the reviewer that ESMs performance offshore, where the regional model boundary is located, is most important for regional downscaling and may differ from those on shelf, although we suspect that performances on the shelf is similar to performance along the boundaries. To address this specific point that was also raised by reviewer 1, we added an analysis of ESM performance along the ACM boundaries and compare those with the results from the shelf (see section 3.2 and supporting Table S2 and Figures S10–S13).

As mentioned in the response to reviewer 1's general comments, we also clarified the objectives and findings in the revised manuscript and now provide further context about the regional use of ESM data. ESM projections are used to drive higher trophic level models and to assess societal impacts of climate change, such as fish catch, that affect mainly Exclusive Economic Zones (EEZ), i.e. coastal ecosystems. Our results indicate that the choice of ESM for these projections is very important. We added the following text on lines 60 to 64:

"*Despite these issues, CMIP historical simulations and future projections have been used to characterize biological responses to climate change in the NWA (e.g., Bryndum-Buchholz et al., 2020a; Greenan et al., 2019; Lavoie et al., 2019; Stortini et al., 2015; Wilson et al., 2019; Wilson and Lotze, 2019). ESM selection in these regional studies is either qualitative or based on either scenario outcomes (e.g. variability across models) or global assessments rather than on regional model performance. However, ESMs that poorly represent the dynamics of the NWA will affect the results of regional studies.*"

**Comment:**
2. They are not discussing the processes that lead to the observed values in the region under study and they are not analysing if the models do represent these processes correctly. I believe salinity should be included, as surface temperature depends strongly on atmospheric forcing while salinity is more representative of the different water masses in that region.

***Response:*** The objectives of our study are: 1) to provide users of ESM output with information about model performance, either for direct use in shelf regions (see previous comment) or for regional downscaling, and to compare with a high-resolution regional model. We now distinguish more clearly between the two types of usage of ESM output. We also discuss some potential sources of mismatch between models and observations but since only limited output is available from the ESMs (e.g., only monthly means of surface properties for CMIP5) we are not able to properly analyze the underlying reasons. We note that such an analysis is not the objective of this study and outside of the intended scope. We now discuss the findings of Lavoie et al (2019) about the parameterization of vertical fluxes and remineralization in the biogeochemical models. For instance, we added the following discussion:

L315-322: "*Lavoie et al. (2019) indicated that the misrepresentation of primary production in the NWA may be associated with the misrepresentation of particulate organic matter sinking and remineralization in the subsurface layer. They found an annual subsurface nitrate peak in CanESM2, GFDL-ESM2M, NorESM1-ME, CESM1-BGC (models 2, 7, 18 and 3, respectively) similar to the high surface nitrate found in this study (supporting Figures S1 and S3). However, all these models had poor scores in our assessment and therefore do not provide an appropriate representation of the biogeochemistry on the NWA shelf (Figure 8) or along the ACM boundaries (Figure S11). However, it is not possible, and beyond the scope of this work, for us to draw conclusions about the source of the regional mismatch in surface chlorophyll and nitrate in the ESMs.*"

L403-407: "*Lavoie et al. (2019) suggested that the PISCES biogeochemical model may underestimate subsurface remineralization in the CNRM and IPSL models, resulting in low surface nutrients where the Gulf Stream detaches from the coast. Our rankings (shelf and offshore) do not support this hypothesis; high surface nitrate concentrations were present in the CNRM models (throughout the region) and the IPSL-CM5A models (around the GoM) (Figures S1–4, S7, S9).*"

We included temperature in the comparison because it is an important variable for higher trophic level studies and climate change impacts. The fact that surface temperature is available at high spatial and temporal resolution on the shelf, similar to chlorophyll, is also important. Despite the tight control by atmospheric forcing, we did find significant differences in surface temperature across the ESMs. We believe these differences are of interest and relevant to many users.

Also, based on the Reviewer's suggestion, we have now included salinity as a fourth variable in our assessment (see Table 2, Figures 4–8, 12), that is discussed throughout the manuscript, such as in Sections 3.1 and 3.2:

L180–183: "*The range of simulated surface salinity is large (Figure 4d–f). Most models overestimate salinity in the GoM (bias = +1.46, Figure 4d). The mismatch is large on the SS and GB but not consistent among models, except for an annual, positive bias in CMIP6 models (bias = +1.42 and +0.76 respectively, Figure 4e–f). In the two latter*

*regions, the biases in CMIP5 models compensate each other, resulting in an ensemble mean close to the observations.*"

L215–217: "*For temperature and salinity, models 3, 20–21, and 24–25 have the largest discrepancy with observations and some clearly represent better the annual cycle than others. The best models for temperature (5–6, 14, 16 28) do not always match the best for salinity (5, 16, 27–28, 30).*"

L226–232: "*As observed previously in Figure 6, the scores of ESMs have a much larger range of variability for temperature (1.5–7.8), salinity (0.5–4.2) and nitrate (1.4–13.2) than for chlorophyll (0.81–1.42) due to the large mismatch observed with a few models (Figure 7, supporting Figures S1–S5). For temperature, 4 of the 6 poorest (largest) scores (> 4.5) are in the CMIP6 group. They all markedly overestimate temperature, especially in the GoM (see supporting Figures S1, S4–S5), except for model 4 that underestimates temperature in SS and GB. The other models have also the poorest scores with respect to salinity. They all largely overestimate salinity in the 3 regions and are clearly outliers with respect to their CMIP category.*"

**Comment:**
3. Moreover, it is very surprising that a similar study (Lavoie et al. 2019) in the exact same area, with the same purpose, and using some of the same ESMs is hardly mentioned at all. No comparison of the results of this study with the 2019 study is made.

*Response:* The study of Lavoie et al (2019) is different in that their main focus is on future projections, but it is carried out in the NWA and is therefore relevant, of course. We now discuss and cite their findings where appropriate and also included a recent report by Lavoie et al (2020) on regional downscaling in the NWA as well as a previous report by Lavoie et al (2013) that provided qualitative comparisons of CMIP5 models with observations.

**Comment:**
4. Also there is not enough details on the comparison with the data, they appear to be comparing different time periods (see detailed comments) or on how the ESMs were brought to a single grid.

*Response:* We provide the information about time range, averaging and spatial mapping in the Methods. Additional information is now added for completeness sake, as detailed in the responses to detailed comments 18–20, 27 and 30 below.

**Comment:**
5. There is only a vague mention of what the improvements are between the CMIP5 and CMIP6 models. What was improved should be stated (not only biogeochemistry of physics) so that the reader can judge on the potential impact on the ranking.

*Response:* See response to comment 2 above and comment 25 below. Model changes from CMIP5 to CMIP6 can be significant, including in the atmospheric and terrestrial realms. This study is not meant to pinpoint the sources of improvement in performances. Given the limited output available from these models, we can only speculate on the

sources of improvement based on our results. The reader is referred to the specific papers listed in Table 1 for the list of changes in the models. To clarify this point, we added the following statement:

*L424-425: "For specific changes in the CMIP6 model versions, the reader is referred to the references listed in Table 1."*

**Comment:**
6. Increasing the model resolution in order to improve the representation of the circulation in the NWA has been mentioned by many authors (e.g. Loder, Brickman, Yool). Here it is stated that the resolution does not have an impact. This is a big statement, considering the general agreement, and it should be demonstrated. The authors could show the changes in circulation of a few models they are giving in example for this.

*Response:* Our findings are in line with previous work, including the ones cited above, see responses to detailed comments 9, 16, and 34–35.

**Comment:**
7. All these points should be addressed in order for the conclusions to be more convincing (ranking based on analysis of shelf surface conditions representative of boundary conditions).

*Response:* These points are addressed in the detailed comments below.

**Comment:**
8. Also, Lavoie et al. (2019) estimated that the boundary conditions obtained with the ESMs were not as reliable for the simulation of the conditions on the Scotian Shelf and in the Gulf of Maine. It would good to know if there was an improvement in this regard with the CMIP6 ESMs.

*Response:* An analysis of model performance along the ACM boundaries is now provided in the revised manuscript. We found similarities in the CMIP5 and CMIP6 rankings for all variable except temperature but with more variability in the ranking of individual variables. This further supports the findings of Lavoie et al (2019). The variability tends to decrease in the CMIP6 models, which suggests more reliability of boundary conditions obtained from the ESMs. However, given the disagreement between shelf and boundary ranking for temperature, we do not feel that we can go further in our discussion of the reliability of boundary conditions than we have.

**Specific comments**

**Comment:**
9. Line 11: Here you say that the coarse resolution is not appropriate to represent the circulation and elemental flux but later on you say that increasing the resolution does not matter. Is it important or not?

*Response:* The two statements are not in contradiction. It is well known that the coarse resolution of ESMs is an issue with regard to resolving shelf-scale processes and that a high resolution is necessary in these areas, as mentioned in comment 6 above. However, even the highest resolution ESM in the ensemble is too coarse to resolve shelf-scale processes and therefore it is not surprising that we do not yet see better performance with increasing ESM resolution. We have now clarified this point by adding the following sentence on Line 392:

*"The lack of correlation between model resolution and performance on the NWA shelf is not surprising as all ESMs are coarse and do not explicitly resolve shelf-scale processes but rather rely on their parameterisation. Much higher resolution will be necessary…"*

**Comment:**
10. Line 14: ability to reproduce surface observations...

*Response:* Done.

**Comment:**
11. Line 15: why is it particularly sensitive?

*Response:* We refer to the effects of climate change on the location and strength of the Gulf Stream and Labrador Sea currents. We modified the sentence as follows:

L13-15: *"The NWA region is biologically productive, influenced by the large-scale Gulf Stream and Labrador Current systems, and particularly sensitive to climatically induced changes in large-scale circulation."*

**Comment:**
12. Line 16: The spatial mismatch in large-scale circulation was not demonstrated. There are references for CMIP5 but what about CMIP6. Changes, or not, in circulation should be shown/mentioned after an inspection of the ESMs results.

*Response:* We mentioned a warm bias in the Gulf of Maine that is in line with the results of Loder et al. (2015) and Saba et al. (2016) (Lines 365-266). Although smaller, a cold bias appears on Grand Banks in most models (Figures 4c and 5c). The biases suggest a mismatch in the large-scale currents. However, since we did not compare the position of the currents across the models, we rephrased the sentence L15-16 as follows:

*"Most ESMs compare relatively poorly to observed nitrate and chlorophyll and show differences with observed temperature and salinity that suggest spatial mismatches in their large-scale current systems."*

With the addition of salinity, we now have more support for this statement:

L305-307: *"A warm bias and a general overestimation of surface salinity in most models indicate a mismatch in the location of the Gulf Stream that influences conditions on the shelf, in line with the previous results of Loder et al. (2015) and Saba et al. (2016)."*

**Comment:**

13. Line22: How can we say just by looking at the surface temperature, nitrate and chl a that the top three models are appropriate for boundary forcing? The model boundaries are hundreds of meters deep (and more) and are not located in the regions analysed. It should be mentioned what are the tracers that will be downscaled at the boundaries? Salinity is certainly one of them, why was is not included in the analysis?

*Response:* The revised manuscript includes salinity and a comparison along the offshore boundaries of the ACM, see responses to comments 2 and 8 above, and response to comment 1 by reviewer 1. Based on the new results we modified our statement as follows:

L22–24: "*An additional evaluation of the ESMs along the regional model boundaries shows larger variability but is generally consistent with the ranking on the shelf. Overall, 11 ESMs were deemed satisfactory for use in the NWA, either directly or for regional downscaling.*"

**Comment:**
14. Main text Line 71: why look only at three variables? What about salinity?

*Response:* Salinity is now included, see response to comment 2 above. Not all variables were available for all models (ESMs and ACM) and could be compared to observations, so we restricted the comparison to 4 variables (with the addition of salinity) in the revised manuscript. The selected variables are, arguably, the most important to potential users.

**Comment:**
15. Line 78: historical simulations are not used for projections. This should be rephrased.

*Response:* The sentence was rephrased as follows:

L85-88: *The comparison provides an overview of ESM performance in the NWA and shows sufficient confidence for only a third of the ESMs. The regional model clearly outperformed all the global models and regional downscaling using single ESM forcing (as opposed to an ensemble) is recommended.*

**Comment:**
16. Lines 115-116: The ESMs horizontal resolution in the region of interest should be given in Table 1. Some models have a variable resolution and it might not be that bad in the NWA.

*Response:* To give a sense of horizontal resolution that is easily comparable across models we have provided the number of grid cells in the three zones of interest in Table 1. This value also depends on the coverage, which can be very poor for coarse grids (e.g. IPSL–CM5), and therefore we believe this provides the most readily useful information to the reader. However, as suggested by the reviewer and since resolution is typically reported in degrees for ESMs, we added a column to Table 1 with average Δlon × Δlat on the NWA shelf. We also added the number of vertical levels.

**Comment:**

17. Line 117: MR and HR mean medium resolution and high resolution respectively. If they share the same grid where does the change in resolution come from?

*Response:* MR stands for Mixed Resolution and HR for Higher Resolution in the MPI model names. MPI-ESM-MR (CMIP5) and MPI-ESM1-2-HR (CMIP6) have the same ocean circulation model, but the horizontal resolution of the atmospheric component was improved from ~200 km (MPI-ESM-MR) to ~100 km (MPI-ESM1-2-HR). Thus, model names are not related to the ocean model, which can be confusing. A similar confusion can occur from the IPSL CMIP5 model names. In this case, the models share the same ocean model, but the horizontal resolution of the atmospheric model is higher in the medium resolution (MR) version compared to the low resolution (LR) version. To avoid the potential for confusion, the following text was added to the caption of Table 1:

*"Note that the IPSL-CM5 models share the same ocean component with a higher resolution atmospheric component in the MR version. Similarly, MPI-ESM-MR and MPI-ESM1-2-HR share the same ocean component with a higher resolution atmospheric component in the HR version."*

**Comment:**
18. Line 123: From where were the satellite data obtained. Who did the averaging?

*Response:* Links to the data were added as follows:

L132-137: *"1) satellite surface chlorophyll observations from the Sea-viewing Wide Field-of-view Sensor (SeaWiFS) as 8-day averaged maps at 1/12° resolution (1999–2010, https://doi.org/data/10.5067/ORBVIEW-2/SEAWIFS/L3M/CHL/2018), 2) surface nitrate from the World Ocean Atlas 2013 (WOA; Garcia et al., 2014) at 1° resolution, 3) daily surface temperature from the Operational SST and Sea Ice Analysis (OSTIA) system (Donlon et al., 2012) at 1/20° resolution (2006–2016, https://doi.org/10.5067/GHOST-4FK01) and 4) surface salinity from the WOA at 1/4° resolution (Zweng et al., 2013). Monthly climatologies were calculated for each of these."*

The following references were added:

SeaWiFS. NASA Goddard Space Flight Center, Ocean Ecology Laboratory, Ocean Biology Processing Group. Sea-viewing Wide Field-of-view Sensor (SeaWiFS) Chlorophyll Data; NASA OB.DAAC, Greenbelt, MD, USA. doi:10.5067/ORBVIEW-2/SEAWIFS/L3M/CHL/2018. Accessed on 2014/03/12.

OSTIA. UK Met Office. 2005. GHRSST Level 4 OSTIA Global Foundation Sea Surface Temperature Analysis. Ver. 1.0. PO.DAAC, CA, USA. doi:10.5067/GHOST-4FK01. Accessed on 2019/12/06.

The original data were daily (OSTIA) and 8-day (SeaWiFS) maps which were converted to monthly climatologies, as mentioned on Lines 133–137.

**Comment:**

19. Line 128: Which data from the AZMP were used? Along the Halifax line only? Why were the data averaged seasonally and not monthly like the other data?

*Response:* See also response to comment 30. Yes, we used data along the Halifax Line where both high-resolution glider data and ship-based bi-monthly or seasonal data are available. The location of the data is presented in Figure 1. The glider missions and the AZMP data collection frequency along the Halifax Line were seasonal, which is why the spatially resolved dataset was averaged into seasons rather than months. This information was added in the paragraph (see above). At station 2, we were able to use a bi-weekly frequency for the AZMP climatology. We rephrased the text in the manuscript as follows for clarity:

L138-143: "*In addition, the regional model was validated using high-resolution in-situ observations along the Halifax Line (Figure 1) from the Atlantic Zone Monitoring Program (AZMP, 2000–2014, http://www.meds-sdmm.dfo-mpo.gc.ca/isdm-gdsi/azmp-pmza/index-eng.html) and glider transects between 2011 and 2016 (Ross et al., 2017). To enable a quantitative comparison between the glider and ACM data (Table 3), we spatially interpolated both datasets onto a transect following the Halifax Line (black line in Figure 1). Glider missions were seasonal and therefore both glider and AZMP transects data were seasonally averaged. For each mission, data were extracted at Station 2 to produce a monthly climatology.*"

**Comment:**
20. Line 132: So the model results are brought back onto 3 different grids, one for each variable. Are the time period also adjusted? For example SST goes from 2006 to 2016. The CMIP5 historical period ends in 2005. How can the two be compared then? Also, there are probably models that have a higher resolution than 1∘ (see my comment for lines 115-116), what is the impact of decreasing the resolution (converting from higher to lower resolution) and having on the ranking analysis. And how is the conversion of one grid to the other done?

*Response:* Since we used a heterogeneous data set, we brought the data and model to the same temporal (monthly) and spatial (observation grid) scale for comparison. We used a long-term climatology for robustness, so that our conclusions aren't affected by interannual variability. Ideally, we would use the same time range for observations and models, but this was not possible. All the ESMs used the same time range (30 years climatology, 1976–2005) so their intercomparison is robust. Note that Line 193 should read "(1976–2005)", not "(1975–2005)", which was corrected in the revised manuscript. Unfortunately, we could only run the ACM simulation for 15 years starting in 1999 so the ESMs and ACM simulations overlap for 6 years only. Since the CMIP6 historical simulations end in 2014 it was possible to use the range 2000–2014 with the CMIP6 models. However, the ESM intercomparison would have been less robust and we decided to use the same time range for all the ESMs. To assess the potential bias associated with the selected time range, we compared the scores of the CMIP6 models when averaged over the years 2000–2014, the same time range as for the regional model and similar to

the observations of chlorophyll and temperature. Except for temperature in model 30, the scores and ranks are consistent with the previous, see Figures R1 and R2 below. Figure R1 is now included in the revised supplement (Figure S14) and we have now added a paragraph on scores uncertainties in the Discussion (L361-374).

[Figure]

*Figure R1. Relationships between scores from CMIP6 model climatologies averaged over 1976–2005 (x-axis) and 2000–2014 (y-axis). ACM (1) is indicated as reference.*

[Figure]

*Figure R2. Relationships between ranks from CMIP6 model climatologies averaged over 1976–2005 (x-axis) and 2000–2014 (y-axis). ACM (1) is indicated as reference.*

The conversion from grid to grid is simply a nearest neighbor interpolation onto the observations grid. This information is added as follows:

L145-146: "*For comparison with the observations, each model was mapped onto the SeaWiFS, WOA and OSTIA grids using a nearest neighbor interpolation.*"

**Comment:**
21. Also, how the thickness of the first grid cell compares between the different models?

*Response:* The ESMs have various vertical resolutions. For completeness, the number of vertical levels was added to Table 1.

**Comment:**
22. Line 175: What is the main difference between the CMIP5 and CMIP6 groups, why is it better? Improved BGC? Same question hold for nitrate.

*Response:* In the discussion, we speculate about the source of improvement in surface chlorophyll and nitrate fields (see section 4.6). The suggested sources of improvement refer to the literature as we cannot substantiate the reasons for these changes from the publicly available data.

**Comment:**
23. Line 200: Figure 6 does not show the annual cycle.

*Response:* Here we refer to the data that are used to calculate the RMSD. The sentence was modified to:

L221-222: "*some models are much better at representing the observed annual cycle (Figure 6), as indicated by the lower RMSD.*".

**Comment:**
24. Line 208: Could you explain why? From local atmospheric forcing or circulation change?

*Response:* We can only speculate about the reasons why some models have poor scores for temperature. In Lines 305–307 we mention the warm bias associated with a mismatch in the location of the Gulf Stream. These models also overestimate salinity, so this explanation is plausible. This information was added as follows:

L229-232: "*They all markedly overestimate temperature, especially in the GoM (see supporting Figures S1, S4–S5), except for model 4 that underestimates temperature in SS and GB. The other models have also the poorest scores with respect to salinity. They all largely overestimate salinity in the 3 regions and are clearly outliers with respect to their CMIP category.*"

However, we do not know why some CMIP6 models (CESM2, GISS-E2-1-G-CC) have a large temperature and salinity bias. These models already had poor scores for temperature and salinity in their CMIP5 version. The following sentence was added:

L417-429: "*Models with poor scores for temperature and salinity (CESM2, GISS-E2-1-G-CC) had already poor scores in their CMIP5 version and therefore the cause of their poor performance is likely the same.*"

**Comment:**
25. Line 209: What are the improvements in the CMIP6 models?

*Response:* Here "improvement" refers to the lower chlorophyll scores for the CMIP6 models 22 and 23 (CNRM-ESM2-1 and GFDL-ESM4). These models have the best chlorophyll scores after ACM. For clarity, "improvement" was removed and the sentence is now:

L232-235: "*The range of variability in chlorophyll scores did not reduce from CMIP5 to CMIP6 and given the relatively low scores of a few CMIP6 models (i.e. 22 and 23), the*

*range is larger in the CMIP6 group (0.8–1.4, Figure 7, right panel) than in the CMIP5 group (1–1.4, Figure 7, left panel).*"

**Comment:**
26. Line 215: So this means that the model ranked 2 (and others) might not be ranked as high? What is the impact on the final choice?

*Response:* The numbers in the parenthesis correspond to the model ID. Since we excluded Nov–Jan from the WOA dataset, i.e. when nitrate is high at the surface, models with consistently low nitrate have lower scores than they would if all winter months were included, and it increases their rank with respect to nitrate. The overall rank is an average of the 4 variables so it should be less sensitive to this effect. Supporting Figures S1-S5j-l indicate that this might be the case for models 4, 8, 14, 19, and 26–27. Models 4, 8, 14 and 19 are part of the outer ensemble so the underestimation of the nitrate score has no effect on the final choice. However, for model 26–27, information on the underestimation of the nitrate scores should be provided. To reflect a possible bias in the ranking of these models we added an asterisk in Table 2 beside the overall rank of these models and the caption was updated accordingly.

**Comment:**
27. Line 218: Could the fact that you are using different time periods and different grid resolution for the three variables explain the lack of correlation?

*Response:* We used climatologies to remove the influence of time on the comparisons as much as possible. The new comparison of the scores calculated over the period 1976-2006 and 2000–2014 shows that time does not influence the scores (see response to Comment 20 above. Correlations are consistent between physical and biological variables, despite the different grid resolutions. Nitrate scores did not correlate with any of the other variables and therefore the source of errors for nitrate seems to differ from that of the other variables.

**Comment:**
28. Line 221: How do you explain that?

*Response:* We do not wish to speculate about the reasons for each models' individual ranking. As stated before, given the limited output that is available for each of the models we would have to guess, but this is outside the intended scope of this study. The objective is to report on the models' behaviour.

**Comment:**
29. Line 230: Why? Does it relate to temperature-dependant phytoplankton growth?

*Response:* Again, we can only speculate. Temperature-dependant phytoplankton growth is a possibility, large scale circulation is another. Except for models 14 and 19, all the models in Group C have poor temperature and salinity scores, and therefore their poor representation of chlorophyll is likely due to a mismatch in large scale circulation. Model 14 has poor salinity and chlorophyll scores but is the best ESM for temperature, whereas model 19 has poor temperature and chlorophyll scores but represents well salinity. Given

the mismatch in error statistics for temperature and salinity in these 2 models, the influence of circulation is probably different. We added the following discussion about the source of mismatches in chlorophyll and nitrate:

L311-322: "*The correlation between temperature and chlorophyll scores (and to a lesser extent salinity) and the concomitant poor scores in chlorophyll and temperature/salinity (i.e. Group C in Figure 7) indicate that errors in surface chlorophyll concentration are partly driven by a misrepresentation of the general circulation and, more generally, of ocean physics. The improvement in chlorophyll from CMIP5 to CMIP6 without an associated improvement in temperature suggest that the errors in surface chlorophyll were also driven to some extent by errors in the biogeochemical model component. Lavoie et al. (2019) indicated that the misrepresentation of primary production in the NWA may be associated with the misrepresentation of particulate organic matter sinking and remineralization in the subsurface layer. They found an annual subsurface nitrate peak in CanESM2, GFDL-ESM2M, NorESM1-ME, CESM1-BGC (models 2, 7, 18 and 3, respectively) similar to the high surface nitrate found in this study (supporting Figures S1 and S3). However, all these models had poor scores in our assessment and therefore do not provide an appropriate representation of the biogeochemistry on the NWA shelf (Figure 8) or along the ACM boundaries (Figure S11). However, it is not possible, and beyond the scope of this work, for us to draw conclusions about the source of the regional mismatch in surface chlorophyll and nitrate in the ESMs.*"

L403-407: "*Lavoie et al. (2019) suggested that the PISCES biogeochemical model may underestimate subsurface remineralization in the CNRM and IPSL models, resulting in low surface nutrients where the Gulf Stream detaches from the coast. Our rankings (shelf and offshore) do not support this hypothesis; high surface nitrate concentrations were present in the CNRM models (throughout the region) and the IPSL-CM5A models (around the GoM) (Figures S1–4, S7, S9).*"

**Comment:**
30. Line 245: What are the years compared for the ACM and the glider data?

*Response:* Information on the ACM and glider data is provided in the Methods. The ACM data are the same as for the comparison with the ESMs, i.e. years 2000–2014 but presented as a seasonal (Figure 9) and daily (Figure 10) climatology to match the resolution of the glider data. The AZMP years are the same as ACM. The glider missions were carried out between 2011 and 2016 but were heterogeneous in time and space (see tracks on Figure 1). To enable a quantitative comparison between the glider and ACM data (Table 3), we spatially interpolated both dataset onto a transect following the Halifax Line (black line in Figure 1). The glider missions were seasonal, which is why the spatially resolved dataset was averaged into seasons (Figure 9). For each mission, data were extracted at Station 2 to produce a monthly climatology (Figure 10). ACM data were extracted at this location for comparison. We added this information as follows in the Methods section:

L140-143: "*To enable a quantitative comparison between the glider and ACM data (Table 3), we spatially interpolated both datasets onto a transect following the Halifax*

*Line (black line in Figure 1). Glider missions were seasonal and therefore both glider and AZMP transects data were seasonally averaged. For each mission, data were extracted at Station 2 to produce a monthly climatology.*"

**Comment:**
31. Line 260: Correlation coefficients are high for nitrate despite having a large bias and RMSD. This should be explained.

*Response:* The correlation coefficient is a complementary measure to bias and RMSD. Correlation and bias are largely unrelated. The former is a measure the similarity in spatial or temporal variations but does not account for bias. In other words, the same correlation coefficient can occur for very different values of bias. Likewise, high correlation does not imply low RMSD. In a noisy data set the RMSD will be higher than in a data set that is smooth, while both might display the same correlation.

**Comment:**
32. Line 270: See my previous comments about time-period and grid differences. I think that a statement about a misrepresentation of ocean physics as the cause should be backed up since later on the cause for nitrate mismatch is stated as coming from the BGC behavior (line 279). There are refs for the CMIP5 models but was there an improvement in circulation with the CMIP6 group or not?

*Response:* Likely, the mismatch is partly associated with ocean physics and partly due to the BGC model. To reflect this, the sentence was modified as follows (see also the response to Comment 29 above):

L311-315: "*The correlation between temperature and chlorophyll scores (and to a lesser extent salinity) and the concomitant poor scores in chlorophyll and temperature/salinity (i.e. Group C in Figure 7) indicate that errors in surface chlorophyll concentration are partly driven by a misrepresentation of the general circulation and, more generally, of ocean physics. The improvement in chlorophyll from CMIP5 to CMIP6 without an associated improvement in temperature suggest that the errors in surface chlorophyll were also driven to some extent by errors in the biogeochemical model component.*"

**Comment:**
33. Line 288: So here again, the model-data comparison was made on a different grid than for the ESMs. Shouldn't it be done on the same grid for an appropriate comparison?

*Response:* No, the grid for comparison depends on the dataset. Here, since we have both high (glider) and low (AZMP) spatial resolution data, we mapped the data along the Halifax Line.

**Comment:**
34. Line 295: Lavoie et al. (2019) also point at the misrepresentation of the remineralisation depth in those models as a likely cause. This also explain why some models having a coarse resolution still have good results with biogeochemistry. But the statement made below that improving the model resolution does not improve the representation of circulation and main features in the models, such as the representation

of the Gulf Stream detachment point and flow around the Grand Banks should be demonstrated. There is a large consensus on that and it should not be stated lightly. The authors could actually show the mean currents between the two versions of a same model with improved resolution. Especially that you state that higher resolution is required to refine the projections on line 306. There is a contradiction here.

*Response:* See also response to comment 9 above. We agree with the Reviewer that there is a large consensus on the effect of grid resolution on large scale circulation and our discussion is in line with this consensus. The resolution of the CMIP models is much coarser than the resolution of the models used to study the effect of grid resolution on the large-scale current systems of the NWA. Therefore, as also pointed out by Reviewer 1 (see comment 7 by Reviewer 1), it is not surprising that current ESMs do not show the effect of grid resolution on model performances; much higher resolution will be necessary to see this effect. We clarified this point as follows after Line 390:

"*The lack of correlation between model resolution and performance on the NWA shelf is not surprising as all ESMs are still coarse and do not explicitly resolve shelf-scale processes but rather rely on their parameterisation. Much higher resolution will be…*"

We also added the following sentence Line 315:

"*Lavoie et al. (2019) indicated that the misrepresentation of primary production in the NWA may be associated with the misrepresentation of particulate organic matter sinking and remineralization in the subsurface layer. They found an annual subsurface nitrate peak in CanESM2, GFDL-ESM2M, NorESM1-ME, CESM1-BGC (models 2, 7, 18 and 3, respectively) similar to the high surface nitrate found in this study (supporting Figures S1 and S3). However, all these models had poor scores in our assessment and therefore do not provide an appropriate representation of the biogeochemistry on the NWA shelf (Figure 8) or along the ACM boundaries (Figure S11).*"

**Comment:**
35. Line 310: Here again it appears to be contradictory as you previously mentioned that BGC improvements we the cause for improvements in the CMIP6 ranking. There are likely different versions of the 4 BGC models mentioned, which should be specified in the table and considered in the analysis. Also, it could relate to the processes that control nitrate in the regions under study, they are different for your 3 regions. And how well are these processes represented by the ESMs?

*Response:* Here we refer to the general BGC component. There are not enough data available to compare specific model versions or parameterizations. This paragraph is meant to point out that, in our comparison, there was no relationship between the type of model and the overall performances. But we cannot go further, and this is not the objective of the study. For clarification the paragraph was modified as follows:

L395-402: "*Although model performance is likely influenced by the biogeochemical model structure, we did not find a clear relationship between the type of biogeochemical model and performance. Here we only refer to the model type because the same model may have different parameterizations when used by different groups. While the inner and*

*outer ensembles share only 3 biogeochemical models (PISCES, HAMOCC, TOPAZ2) out of 13, there was no indication of consistently better performance for the biogeochemical models in the inner ensemble. For example, models using similar ocean biogeochemistry (e.g., PISCES: 5, 12–14 (CMIP5), 22 and 26 (CMIP6), and HAMOCC: 15–16, 18 (CMIP5), 28–29 (CMIP6)) had very different ranks, with no obvious relationship between overall model rank and the ocean biogeochemical model component. Moreover, 5 and 4 biogeochemical models were represented in the 5 best ranked ESMs on the NWA shelf and outer ACM boundaries, respectively, similar to previous findings by Rickard et al. (2016). Lavoie et al. (2019) suggested that the PISCES biogeochemical model may underestimate subsurface remineralization in the CNRM and IPSL models, resulting in low surface nutrients where the Gulf Stream detaches from the coast. Our rankings (shelf and offshore) do not support this hypothesis; high surface nitrate concentrations were present in the CNRM models (throughout the region) and the IPSL-CM5A models (around the GoM) (Figures S1–4, S7, S9).*"

**Comment:**
36. Line 335: What was updated in the ocean biogeochemistry?

*Response:* As mentioned in the response to previous comments above, our goal is not to discuss the details of the models. The HAMOCC biogeochemistry module includes a new parameterization of detritus sinking, which may influence surface chlorophyll and nitrate, as suggested by Lavoie et al (2019). However, this explanation is speculative and we do not think that it should be included here.

**Comment:**
37. Figure 4f: In the suppl. figures, there is more chl a in the model than in the obs. The opposite is shown here.

*Response:* The supplemental figures S1–S5 present the individual chlorophyll time series for the 29 ESMs, whereas Figure 4f shows ECM ensembles. The "all" ensemble time series are calculated with all the individual ESM time series in Figures S1–S5. Figure 4f shows that even though individual ESMs can be close to observations during the spring bloom (e.g. HadGEM2, Figure S2f) and even significantly larger (e.g. CESM2, Figure S4f), the ensemble of all ESMs underestimates the bloom.

**Comment:**
38. Figure 7: Maybe use ACM instead of ROMS.

*Response:* Done.

**Comment:**
39. Figure 8: Could specify that ACM has the same rank for the three variables (only see one point)

*Response:* The following sentence was added to the figure caption:

[revised manuscript text omitted]

Formatted Table ... [2]
Merged Cells ... [3]
Merged Cells ... [4]
Split Cells ... [6]
Inserted Cells ... [5]
Inserted Cells ... [9]
Formatted Table ... [7]
Inserted Cells ... [8]
Inserted Cells ... [10]
Formatted ... [11]
Merged Cells ... [12]
Inserted Cells ... [15]
Formatted Table ... [13]
Inserted Cells ... [14]
Split Cells ... [16]
Merged Cells ... [17]
Formatted ... [18]
Inserted Cells ... [19]
Inserted Cells ... [20]
Merged Cells ... [21]
Inserted Cells ... [22]
Inserted Cells ... [23]
Merged Cells ... [24]
Inserted Cells ... [27]
Formatted Table ... [25]
Inserted Cells ... [26]
Split Cells ... [28]
Merged Cells ... [29]
Formatted ... [30]
Inserted Cells ... [32]
Formatted Table ... [31]
Inserted Cells ... [33]
Merged Cells ... [34]
Inserted Cells ... [37]
Formatted Table ... [35]
Inserted Cells ... [36]
Formatted ... [38]

[revised manuscript text omitted]

**Page 25: [1] Deleted**        Author

**Page 25: [2] Formatted Table**        Author

Formatted Table

**Page 25: [3] Merged Cells**        Author        15/30/07 8:48:00 PM

Merged Cells

**Page 25: [4] Merged Cells**        Author        15/30/07 8:48:00 PM

Merged Cells

**Page 25: [5] Inserted Cells**        Author        15/30/07 8:48:00 PM

Inserted Cells

**Page 25: [6] Split Cells**        Author        15/30/07 8:48:00 PM

Split Cells

**Page 25: [7] Formatted Table**        Author

Formatted Table

**Page 25: [8] Inserted Cells**        Author        15/30/07 8:48:00 PM

Inserted Cells

**Page 25: [9] Inserted Cells**        Author        15/30/07 8:48:00 PM

Inserted Cells

**Page 25: [10] Inserted Cells**        Author        15/30/07 8:48:00 PM

Inserted Cells

**Page 25: [11] Formatted**        Author

Centered

**Page 25: [12] Merged Cells**        Author        15/30/07 8:48:00 PM

Merged Cells

**Page 25: [13] Formatted Table**        Author

Formatted Table

**Page 25: [14] Inserted Cells**        Author        15/30/07 8:48:00 PM

Inserted Cells

**Page 25: [15] Inserted Cells**        Author        15/30/07 8:48:00 PM

Inserted Cells

**Page 25: [16] Split Cells**        Author        15/30/07 8:48:00 PM

Split Cells

**Page 25: [18] Formatted**        Author

Centered

**Page 25: [19] Inserted Cells**        Author        15/30/07 8:48:00 PM

Inserted Cells

**Page 25: [20] Inserted Cells**        Author        15/30/07 8:48:00 PM

Inserted Cells

**Page 25: [21] Merged Cells**        Author        15/30/07 8:48:00 PM

Merged Cells

**Page 25: [22] Inserted Cells**        Author        15/30/07 8:48:00 PM

Inserted Cells

**Page 25: [23] Inserted Cells**        Author        15/30/07 8:48:00 PM

Inserted Cells

**Page 25: [24] Merged Cells**        Author        15/30/07 8:48:00 PM

Merged Cells

**Page 25: [25] Formatted Table**        Author

Formatted Table

**Page 25: [26] Inserted Cells**        Author        15/30/07 8:48:00 PM

Inserted Cells

**Page 25: [27] Inserted Cells**        Author        15/30/07 8:48:00 PM

Inserted Cells

**Page 25: [28] Split Cells**        Author        15/30/07 8:48:00 PM

Split Cells

**Page 25: [29] Merged Cells**        Author        15/30/07 8:48:00 PM

Merged Cells

**Page 25: [30] Formatted**        Author

Centered

**Page 25: [31] Formatted Table**        Author

Formatted Table

**Page 25: [32] Inserted Cells**        Author        15/30/07 8:48:00 PM

Inserted Cells

**Page 25: [33] Inserted Cells**        Author        15/30/07 8:48:00 PM

Inserted Cells

Merged Cells

| Page 25: [35] Formatted Table | Author | |
| --- | --- | --- |

Formatted Table

| Page 25: [36] Inserted Cells | Author | 15/30/07 8:48:00 PM |
| --- | --- | --- |

Inserted Cells

| Page 25: [37] Inserted Cells | Author | 15/30/07 8:48:00 PM |
| --- | --- | --- |

Inserted Cells

| Page 25: [38] Formatted | Author | |
| --- | --- | --- |

Centered

| Page 25: [39] Merged Cells | Author | 15/30/07 8:48:00 PM |
| --- | --- | --- |

Merged Cells

| Page 25: [40] Formatted Table | Author | |
| --- | --- | --- |

Formatted Table

| Page 25: [41] Inserted Cells | Author | 15/30/07 8:48:00 PM |
| --- | --- | --- |

Inserted Cells

| Page 25: [42] Inserted Cells | Author | 15/30/07 8:48:00 PM |
| --- | --- | --- |

Inserted Cells

| Page 26: [43] Formatted | Author | |
| --- | --- | --- |

Line spacing:  single

| Page 26: [44] Formatted Table | Author | |
| --- | --- | --- |

Formatted Table

| Page 26: [45] Inserted Cells | Author | 15/30/07 8:48:00 PM |
| --- | --- | --- |

Inserted Cells

| Page 26: [46] Inserted Cells | Author | 15/30/07 8:48:00 PM |
| --- | --- | --- |

Inserted Cells

| Page 26: [47] Formatted | Author | |
| --- | --- | --- |

Font color: Auto

| Page 26: [48] Formatted | Author | |
| --- | --- | --- |

Font color: Auto

| Page 26: [49] Formatted | Author | |
| --- | --- | --- |

Font color: Auto

**Page 26: [51] Formatted**          **Author**

Font color: Auto

**Page 26: [51] Formatted**          **Author**

Font color: Auto

**Page 26: [52] Formatted**          **Author**

Font color: Auto

**Page 26: [53] Inserted Cells**          **Author**          **15/30/07 8:48:00 PM**

Inserted Cells

**Page 26: [54] Formatted**          **Author**

Font color: Auto

**Page 26: [55] Formatted**          **Author**

Font color: Auto

**Page 26: [56] Formatted**          **Author**

Left

**Page 26: [57] Formatted**          **Author**

Font color: Auto

**Page 26: [58] Formatted**          **Author**

Font color: Auto

**Page 26: [59] Formatted**          **Author**

Font color: Auto

**Page 26: [60] Formatted**          **Author**

Font color: Auto

**Page 26: [61] Inserted Cells**          **Author**          **15/30/07 8:48:00 PM**

Inserted Cells

**Page 26: [62] Formatted**          **Author**

Font color: Auto

**Page 26: [63] Formatted**          **Author**

Font color: Auto

**Page 26: [64] Formatted**          **Author**

Font color: Auto

**Page 26: [65] Formatted**          **Author**

Font color: Auto

**Page 26: [67] Deleted Cells**      **Author**      **15/30/07 8:48:00 PM**

Deleted Cells

**Page 26: [68] Inserted Cells**      **Author**      **15/30/07 8:48:00 PM**

Inserted Cells

**Page 26: [69] Formatted**      **Author**

Font color: Auto

**Page 26: [70] Formatted**      **Author**

Font color: Auto

**Page 26: [71] Formatted**      **Author**

Font color: Auto

**Page 26: [72] Deleted Cells**      **Author**      **15/30/07 8:48:00 PM**

Deleted Cells

**Page 26: [73] Formatted**      **Author**

Font color: Auto

**Page 26: [74] Formatted**      **Author**

Font color: Black

**Page 26: [75] Formatted**      **Author**

Font color: Auto

**Page 26: [76] Inserted Cells**      **Author**      **15/30/07 8:48:00 PM**

Inserted Cells

**Page 26: [77] Inserted Cells**      **Author**      **15/30/07 8:48:00 PM**

Inserted Cells

**Page 26: [78] Inserted Cells**      **Author**      **15/30/07 8:48:00 PM**

Inserted Cells

**Page 26: [79] Formatted**      **Author**

Left

**Page 26: [80] Formatted**      **Author**

Font color: Auto

**Page 26: [81] Inserted Cells**      **Author**      **15/30/07 8:48:00 PM**

Inserted Cells

**Page 26: [82] Formatted**      **Author**

Font color: Auto

Font color: Auto

**Page 26: [84] Formatted**                    **Author**

Font color: Auto

**Page 26: [85] Inserted Cells**               **Author**                    **15/30/07 8:48:00 PM**

Inserted Cells

**Page 26: [86] Formatted**                    **Author**

Font color: Auto

**Page 26: [87] Formatted**                    **Author**

Font color: Auto

**Page 26: [88] Formatted**                    **Author**

Font color: Auto

**Page 26: [89] Formatted**                    **Author**

Left

**Page 26: [90] Formatted**                    **Author**

Font color: Auto

**Page 26: [91] Formatted**                    **Author**

Font color: Auto

**Page 26: [92] Formatted**                    **Author**

Font color: Auto

**Page 26: [93] Formatted**                    **Author**

Font color: Auto

**Page 26: [93] Formatted**                    **Author**

Font color: Auto

**Page 26: [94] Inserted Cells**               **Author**                    **15/30/07 8:48:00 PM**

Inserted Cells

**Page 26: [95] Inserted Cells**               **Author**                    **15/30/07 8:48:00 PM**

Inserted Cells

**Page 26: [96] Formatted**                    **Author**

Font color: Auto

**Page 26: [97] Formatted**                    **Author**

Font color: Auto

**Page 26: [99] Formatted**       **Author**

Font color: Auto

**Page 26: [100] Formatted**       **Author**

Font color: Auto

**Page 26: [101] Formatted**       **Author**

Font color: Black

**Page 26: [102] Formatted**       **Author**

Font color: Auto

**Page 26: [103] Formatted**       **Author**

Font color: Auto

**Page 26: [104] Formatted**       **Author**

Left

**Page 26: [105] Formatted**       **Author**

Font color: Auto

**Page 26: [106] Deleted Cells**       **Author**       **15/30/07 8:48:00 PM**

Deleted Cells

**Page 26: [107] Formatted**       **Author**

Font color: Auto

**Page 26: [108] Formatted**       **Author**

Font color: Auto

**Page 26: [109] Inserted Cells**       **Author**       **15/30/07 8:48:00 PM**

Inserted Cells

**Page 26: [110] Formatted**       **Author**

Font color: Auto

**Page 26: [110] Formatted**       **Author**

Font color: Auto

**Page 26: [111] Formatted**       **Author**

Font color: Auto

**Page 26: [112] Formatted**       **Author**

Font color: Auto

**Page 26: [113] Formatted**       **Author**

Left

**Page 26: [115] Formatted**                                    **Author**

Font color: Auto

**Page 26: [116] Formatted**                                    **Author**

Font color: Auto

**Page 26: [117] Formatted**                                    **Author**

Font color: Black

**Page 26: [118] Formatted**                                    **Author**

Font color: Auto

**Page 26: [119] Formatted**                                    **Author**

Font color: Auto

**Page 26: [120] Formatted**                                    **Author**

Left

**Page 26: [121] Formatted**                                    **Author**

Font color: Auto

**Page 26: [122] Inserted Cells**                   **Author**                      **15/30/07 8:48:00 PM**

Inserted Cells

**Page 26: [123] Inserted Cells**                   **Author**                      **15/30/07 8:48:00 PM**

Inserted Cells

**Page 26: [124] Inserted Cells**                   **Author**                      **15/30/07 8:48:00 PM**

Inserted Cells

**Page 26: [125] Formatted**                                    **Author**

Font color: Auto

**Page 26: [126] Formatted**                                    **Author**

Font color: Black

**Page 26: [127] Formatted**                                    **Author**

Font color: Auto

**Page 26: [128] Deleted Cells**                    **Author**                      **15/30/07 8:48:00 PM**

Deleted Cells

**Page 26: [129] Formatted**                                    **Author**

Font color: Auto

**Page 26: [130] Formatted**                                    **Author**

Font color: Black

Deleted Cells

| Page 26: [132] Formatted | Author | |
Left

| Page 26: [133] Formatted | Author | |
Font color: Auto

| Page 26: [134] Deleted Cells | Author | 15/30/07 8:48:00 PM |
Deleted Cells

| Page 26: [135] Deleted Cells | Author | 15/30/07 8:48:00 PM |
Deleted Cells

| Page 26: [136] Formatted | Author | |
Font color: Auto

| Page 26: [137] Inserted Cells | Author | 15/30/07 8:48:00 PM |
Inserted Cells

| Page 26: [138] Formatted | Author | |
Font color: Auto

| Page 26: [138] Formatted | Author | |
Font color: Auto

| Page 26: [139] Inserted Cells | Author | 15/30/07 8:48:00 PM |
Inserted Cells

| Page 26: [140] Formatted | Author | |
Font color: Auto

| Page 26: [141] Formatted | Author | |
Font color: Auto

| Page 26: [142] Formatted | Author | |
Left

| Page 26: [143] Formatted | Author | |
Font color: Auto

| Page 26: [144] Formatted | Author | |
Font color: Auto

| Page 26: [145] Inserted Cells | Author | 15/30/07 8:48:00 PM |
Inserted Cells

**Page 26: [147] Inserted Cells**          **Author**          **15/30/07 8:48:00 PM**

Inserted Cells

**Page 26: [148] Formatted**          **Author**

Font color: Auto

**Page 26: [149] Deleted Cells**          **Author**          **15/30/07 8:48:00 PM**

Deleted Cells

**Page 26: [150] Formatted**          **Author**

Font color: Auto

**Page 26: [151] Formatted**          **Author**

Font color: Auto

**Page 26: [152] Formatted**          **Author**

Left

**Page 26: [153] Formatted**          **Author**

Font color: Auto

**Page 26: [154] Formatted**          **Author**

Font color: Auto

**Page 26: [155] Formatted**          **Author**

Font color: Auto

**Page 26: [156] Inserted Cells**          **Author**          **15/30/07 8:48:00 PM**

Inserted Cells

**Page 26: [157] Inserted Cells**          **Author**          **15/30/07 8:48:00 PM**

Inserted Cells

**Page 26: [158] Formatted**          **Author**

Font color: Auto

**Page 26: [159] Formatted**          **Author**

Font color: Auto

**Page 26: [160] Formatted**          **Author**

Font color: Black

**Page 26: [161] Deleted Cells**          **Author**          **15/30/07 8:48:00 PM**

Deleted Cells

**Page 26: [162] Formatted**          **Author**

Left

**Page 26: [164] Inserted Cells**      **Author**      **15/30/07 8:48:00 PM**

Inserted Cells

**Page 26: [165] Inserted Cells**      **Author**      **15/30/07 8:48:00 PM**

Inserted Cells

**Page 26: [166] Formatted**      **Author**

Font color: Auto

**Page 26: [166] Formatted**      **Author**

Font color: Auto

**Page 26: [167] Formatted**      **Author**

Font color: Auto

**Page 26: [168] Formatted**      **Author**

Font color: Auto

**Page 26: [169] Formatted**      **Author**

Left

**Page 26: [170] Formatted**      **Author**

Font color: Auto

**Page 26: [171] Deleted Cells**      **Author**      **15/30/07 8:48:00 PM**

Deleted Cells

**Page 26: [172] Deleted Cells**      **Author**      **15/30/07 8:48:00 PM**

Deleted Cells

**Page 26: [173] Formatted**      **Author**

Font color: Auto

**Page 26: [174] Inserted Cells**      **Author**      **15/30/07 8:48:00 PM**

Inserted Cells

**Page 26: [175] Inserted Cells**      **Author**      **15/30/07 8:48:00 PM**

Inserted Cells

**Page 26: [176] Inserted Cells**      **Author**      **15/30/07 8:48:00 PM**

Inserted Cells

**Page 26: [177] Formatted**      **Author**

Font color: Auto

**Page 26: [178] Formatted**      **Author**

Font color: Auto

Left

**Page 26: [180] Formatted**                    **Author**

Font color: Auto

**Page 26: [181] Formatted**                    **Author**

Font color: Auto

**Page 26: [182] Formatted**                    **Author**

Font color: Black

**Page 26: [183] Formatted**                    **Author**

Font color: Auto

**Page 26: [184] Inserted Cells**        **Author**                **15/30/07 8:48:00 PM**

Inserted Cells

**Page 26: [185] Inserted Cells**        **Author**                **15/30/07 8:48:00 PM**

Inserted Cells

**Page 26: [186] Formatted**                    **Author**

Left

**Page 26: [187] Formatted**                    **Author**

Font color: Auto

**Page 26: [188] Inserted Cells**        **Author**                **15/30/07 8:48:00 PM**

Inserted Cells

**Page 26: [189] Inserted Cells**        **Author**                **15/30/07 8:48:00 PM**

Inserted Cells

**Page 26: [190] Formatted**                    **Author**

Font color: Auto

**Page 26: [191] Formatted**                    **Author**

Font color: Auto

**Page 26: [192] Formatted**                    **Author**

Left

**Page 26: [193] Formatted**                    **Author**

Font color: Auto

**Page 26: [194] Formatted**                    **Author**

Font color: Auto

**Page 26: [196] Formatted**      **Author**

Font color: Black

**Page 26: [197] Inserted Cells**      **Author**      **15/30/07 8:48:00 PM**

Inserted Cells

**Page 26: [198] Inserted Cells**      **Author**      **15/30/07 8:48:00 PM**

Inserted Cells

**Page 26: [199] Formatted**      **Author**

Font color: Auto

**Page 26: [200] Formatted**      **Author**

Left

**Page 26: [201] Formatted**      **Author**

Font color: Auto

**Page 26: [202] Inserted Cells**      **Author**      **15/30/07 8:48:00 PM**

Inserted Cells

**Page 26: [203] Inserted Cells**      **Author**      **15/30/07 8:48:00 PM**

Inserted Cells

**Page 26: [204] Inserted Cells**      **Author**      **15/30/07 8:48:00 PM**

Inserted Cells

**Page 26: [205] Inserted Cells**      **Author**      **15/30/07 8:48:00 PM**

Inserted Cells

**Page 26: [206] Inserted Cells**      **Author**      **15/30/07 8:48:00 PM**

Inserted Cells

**Page 26: [207] Formatted**      **Author**

Font color: Auto

**Page 26: [208] Deleted Cells**      **Author**      **15/30/07 8:48:00 PM**

Deleted Cells

**Page 26: [209] Formatted**      **Author**

Font color: Auto

**Page 26: [210] Formatted**      **Author**

Font color: Black

**Page 26: [211] Deleted Cells**      **Author**      **15/30/07 8:48:00 PM**

Deleted Cells

**Page 26: [213] Formatted**         **Author**

Left

**Page 26: [214] Formatted**         **Author**

Font color: Auto

**Page 26: [215] Formatted**         **Author**

Font color: Auto

**Page 26: [216] Formatted**         **Author**

Font color: Black

**Page 26: [217] Inserted Cells**         **Author**         **15/30/07 8:48:00 PM**

Inserted Cells

**Page 26: [218] Formatted**         **Author**

Font color: Auto

**Page 26: [219] Formatted**         **Author**

Font color: Auto

**Page 26: [220] Formatted**         **Author**

Font color: Black

**Page 26: [221] Inserted Cells**         **Author**         **15/30/07 8:48:00 PM**

Inserted Cells

**Page 27: [222] Formatted**         **Author**

Font color: Black

**Page 27: [223] Inserted Cells**         **Author**         **15/30/07 8:48:00 PM**

Inserted Cells

**Page 27: [224] Formatted**         **Author**

Font color: Auto

**Page 27: [225] Formatted**         **Author**

Left

**Page 27: [226] Formatted**         **Author**

Font color: Auto

**Page 27: [227] Inserted Cells**         **Author**         **15/30/07 8:48:00 PM**

Inserted Cells

**Page 27: [228] Inserted Cells**         **Author**         **15/30/07 8:48:00 PM**

Inserted Cells

Font color: Auto

**Page 27: [230] Formatted**                  **Author**
Font color: Black

**Page 27: [231] Formatted**                  **Author**
Font color: Auto

**Page 27: [232] Formatted**                  **Author**
Font color: Auto

**Page 27: [233] Formatted**                  **Author**
Left

**Page 27: [234] Formatted**                  **Author**
Font color: Auto

**Page 27: [235] Inserted Cells**             **Author**              **15/30/07 8:48:00 PM**
Inserted Cells

**Page 27: [236] Inserted Cells**             **Author**              **15/30/07 8:48:00 PM**
Inserted Cells

**Page 27: [237] Formatted**                  **Author**
Font color: Auto

**Page 27: [238] Formatted**                  **Author**
Font color: Auto

**Page 27: [239] Deleted Cells**              **Author**              **15/30/07 8:48:00 PM**
Deleted Cells

**Page 27: [240] Formatted**                  **Author**
Font color: Auto

**Page 27: [241] Formatted**                  **Author**
Left

**Page 27: [242] Formatted**                  **Author**
Font color: Auto

**Page 27: [243] Inserted Cells**             **Author**              **15/30/07 8:48:00 PM**
Inserted Cells

**Page 27: [244] Inserted Cells**             **Author**              **15/30/07 8:48:00 PM**
Inserted Cells

**Page 27: [246] Formatted**        **Author**

Font color: Auto

**Page 27: [247] Formatted**        **Author**

Font color: Black

**Page 27: [248] Deleted Cells**    **Author**        **15/30/07 8:48:00 PM**

Deleted Cells

**Page 27: [249] Deleted Cells**    **Author**        **15/30/07 8:48:00 PM**

Deleted Cells

**Page 27: [250] Deleted Cells**    **Author**        **15/30/07 8:48:00 PM**

Deleted Cells

**Page 27: [251] Formatted**        **Author**

Left

**Page 27: [252] Inserted Cells**   **Author**        **15/30/07 8:48:00 PM**

Inserted Cells

**Page 27: [253] Formatted**        **Author**

Font color: Auto

**Page 27: [254] Formatted**        **Author**

Font color: Auto

**Page 27: [255] Formatted**        **Author**

Font color: Black

**Page 27: [256] Inserted Cells**   **Author**        **15/30/07 8:48:00 PM**

Inserted Cells

**Page 27: [257] Inserted Cells**   **Author**        **15/30/07 8:48:00 PM**

Inserted Cells

**Page 27: [258] Formatted**        **Author**

Font color: Auto

**Page 27: [259] Formatted**        **Author**

Font color: Black

**Page 27: [260] Formatted**        **Author**

Left

**Page 27: [261] Formatted**        **Author**

Font color: Auto

**Page 27: [263] Formatted**            **Author**

Font color: Auto

**Page 27: [264] Formatted**            **Author**

Font color: Auto

**Page 27: [265] Inserted Cells**            **Author**            **15/30/07 8:48:00 PM**

Inserted Cells

**Page 27: [266] Formatted**            **Author**

Font color: Auto

**Page 27: [267] Formatted**            **Author**

Font color: Auto

**Page 27: [268] Formatted**            **Author**

Font color: Auto

**Page 27: [269] Formatted**            **Author**

Font color: Auto

**Page 27: [270] Formatted**            **Author**

Left

**Page 27: [271] Formatted**            **Author**

Font color: Auto

**Page 27: [272] Formatted**            **Author**

Font color: Auto

**Page 27: [273] Formatted**            **Author**

Font color: Auto

**Page 27: [274] Formatted**            **Author**

Font color: Auto

**Page 27: [275] Formatted**            **Author**

Font color: Auto

**Page 27: [276] Formatted**            **Author**

Font color: Auto

**Page 27: [277] Formatted**            **Author**

Font color: Auto

**Page 27: [278] Formatted**            **Author**

Font color: Auto

Left

**Page 27: [280] Formatted**                                    **Author**

Font color: Auto

**Page 27: [281] Formatted**                                    **Author**

Font color: Auto

**Page 27: [282] Formatted**                                    **Author**

Font color: Auto

**Page 27: [283] Formatted**                                    **Author**

Font color: Auto

**Page 27: [284] Formatted**                                    **Author**

Font color: Auto

**Page 27: [285] Formatted**                                    **Author**

Font color: Auto

**Page 27: [286] Formatted**                                    **Author**

Font color: Auto

**Page 27: [287] Formatted**                                    **Author**

Font color: Auto

**Page 27: [288] Formatted**                                    **Author**

Left

**Page 27: [289] Formatted**                                    **Author**

Font color: Auto

**Page 27: [290] Formatted**                                    **Author**

Font color: Auto

**Page 27: [291] Formatted**                                    **Author**

Font color: Auto

**Page 27: [292] Formatted**                                    **Author**

Font color: Auto

**Page 27: [293] Formatted**                                    **Author**

Font color: Auto

**Page 27: [294] Formatted**                                    **Author**

Font color: Auto

**Page 27: [296] Formatted**          **Author**

Font color: Auto

**Page 27: [297] Formatted**          **Author**

Left

**Page 27: [298] Formatted**          **Author**

Font color: Auto

**Page 27: [299] Formatted**          **Author**

Font color: Auto

**Page 27: [300] Formatted**          **Author**

Font color: Auto

**Page 27: [301] Formatted**          **Author**

Font color: Auto

**Page 27: [302] Formatted**          **Author**

Font color: Auto

**Page 27: [303] Formatted**          **Author**

Font color: Auto

**Page 27: [304] Formatted**          **Author**

Font color: Auto

**Page 27: [305] Formatted**          **Author**

Font color: Auto

**Page 27: [306] Formatted**          **Author**

Left

**Page 27: [307] Formatted**          **Author**

Font color: Auto

**Page 27: [308] Formatted**          **Author**

Font color: Auto

**Page 27: [309] Formatted**          **Author**

Font color: Auto

**Page 27: [310] Formatted**          **Author**

Font color: Auto

**Page 27: [311] Formatted**          **Author**

Font color: Auto

**Page 27: [313] Formatted**                         **Author**

Font color: Auto

**Page 27: [314] Formatted**                         **Author**

Font color: Auto

**Page 27: [315] Formatted**                         **Author**

Left

**Page 27: [316] Formatted**                         **Author**

Font: Not Italic

---

## Author Response (AR2)

**Detailed responses to reviewer 2** (reviewer comments are included in black, responses in blue font)

**General comments**

The authors did answer the concerns initially raised. The objectives of the paper were better defined and reflect the work presented. It is a valuable piece of work and it will be useful for scientists seeking to use projections from ESMs as part of their work. I thus recommend that this paper be accepted for publication after some very minor corrections/improvements. See comments below.

*Response:* Thank you for the recommendation and your suggestions (see below).

**Specific comments**

**Comment:**

1. Line 15. I would repeat surface nitrate. Same for observed surface temperature.

*Response:* Done.

**Comment:**

2. Line 52: Maybe you wanted to refer to Gilbert et al. 2005?

*Response:* The reference was added. Both references are relevant, so we also left Gilbert et al. 2010.

**Comment:**

3. Lines 102 and 155: and salinity.

*Response:* Done.

**Comment:**

4. Line 314. Improvement in terms of scores? Could you specify what the improvement is? I see Chl a going from 0.9 to 1.4 to 0.8 to 1.4. That is not a big change. Nitrate improves a lot though.

*Response:* Here the improvement refers so the ranks of specific models. Since this is discussed below in section 4.6, we clarified the sentence as follows:

P12L4-5: "*The improvement in chlorophyll from CMIP5 to CMIP6 in some models without an associated improvement in temperature (see below) suggest that…*"

**Comment:**

5. Line 400. In fact the nutrients that are upwelled at the GS detachment point do not affect directly the Scotian Shelf. These waters are transported towards the sub-arctic gyre. However, mixing with Labrador Sea water occurs south of the Grand Banks and then flows south towards the Scotian Shelf. This is reflected in your better correlation of SS conditions with conditions at the eastern boundary. So you need to have a good correspondence between circulation, water mass representation and biogeochemical conditions. What Lavoie et al. mention is that these models do not represent the observed subsurface maximum. Their figure 13 actually shows that the surface nitrate is high along the whole transect with CNRM-CM5. We can also see in your figure S1 that the nitrate stays high everywhere all year round with this model. Something is clearly not working properly. Your figures S2 and S3 also show that nutrients are low in the CMIP5 IPSL models, except in S2 in the GoM.

*Response:* As indicated by the reviewer, CNRM-CM5 and CMIP5 IPSL models have high and low surface nutrients, respectively. These large scale patterns do not seem to be associated with upwelled GS waters and therefore our results do not fully support the underestimation of subsurface remineralization proposed by Lavoie et al (2019). We clarified the paragraph as follows:

P14L47-59: "*Lavoie et al. (2019) suggested that the PISCES biogeochemical model may underestimate subsurface remineralization in the CNRM and IPSL models, resulting in low surface nutrients where the Gulf Stream detaches from the coast. Our rankings (shelf and offshore) and the spatial patterns in Figures S1- 9 do not fully support this hypothesis; high surface nitrate concentrations were present in the CNRM models throughout the region, whereas concentrations in the IPSL- CM5A models were low (except around the GoM in Spring) (Figures S1–4, S7, S9). It is unlikely that these large-scale patterns are driven by upwelled Gulf Stream waters, although differences in remineralization could influence these general patterns.*"